# Eoarchean and Hadean melts reveal arc-like trace element and isotopic signatures

Wriju Chowdhury [1] ✉, Dustin Trail [1], Martha Miller[1] & Paul Savage[2]

Constraining the lithological diversity and tectonics of the earliest Earth is critical to understanding our planet's evolution. Here we use detrital Jack Hills zircon (3.7 – 4.2 Ga) analyses coupled with new experimental partitioning data to model the silica content, Si+O isotopic composition, and trace element contents of their parent melts. Comparing our derived Jack Hills zircons' parent melt Si+O isotopic compositions ($-1.92 \leq \delta^{30}Si_{NBS28} \leq 0.53$ ‰; $5.23 \leq \delta^{18}O_{VSMOW} \leq 9.00$ ‰) to younger crustal lithologies, we conclude that the chemistry of the parent melts was influenced by the assimilation of terrigenous sediments, serpentinites, cherts, and silicified basalts, followed by igneous differentiation, leading to the formation of intermediate to felsic melts in the early Earth. Trace element measurements also show that the formational regime had an arc-like chemistry, implying the presence of mobile-lid tectonics in the Hadean. Finally, we propose that these continental-crust forming processes operated uniformly from 4.2 to at least 3.7 Ga.

The paucity of terrestrial geological samples from the Hadean and Eoarchaean means that making direct inferences about the geochemical and geodynamic nature of the first 500 Myr of our planet is speculative and often divisive. Many researchers have made implicative arguments about the Hadean through the chemical and isotopic analysis of Hadean zircons and their mineral inclusions. This includes inferences about the types of geodynamic regimes and geological processes that could generate silicic melts (i.e., the earliest continental crust), hydrosphere-lithosphere interactions, and possible mobile-lid tectonics during a time when life may have evolved on our planet[1–7]. Even with access to ancient zircons, the presence or absence of mobile lid tectonics in the Hadean remains a vexed question, with arguments posited to support both end-members. While many have recently proposed a stagnant-lid for the Hadean Earth based on Hf, O, and Nd isotopes[8–11], there are compelling arguments based on geophysical modeling, Nd isotopic evolution models, trace element chemistry of Hadean zircons that point toward a mobile-lid regime on the early Earth[6,12–16]. Ultimately, these end-member models are based on contradictory, but empirical, observations need to be reconciled to provide an accurate view of early Earth's geological history.

We use in-situ analyses of Jack Hills Zircons (JHZs), new zircon-melt partition coefficients, and zircon-melt Si+O isotope fractionation factors to place quantitative constraints on the chemistry of the zircons' parental magmas. The inversion of zircon chemistry to derive parent melt chemistry follows the methodology developed by[6], and we further expand upon that technique by utilizing a new suite of partition coefficients to quantify trace element (TE) and rare-earth (REE) element concentrations, and $SiO_2$ content of the zircon parental melts. Using this information, we then derive the Si+O isotopic values of the Eoarchaean and Hadean melts as well as their formational regimes. Quantifying $SiO_2$ content and Si+O isotopes is especially important given that Si and O constitute ~75 wt.% of rock-forming elements in the crust. The degree of polymerisation of Si-O bonds also correlates with the relative density of the continental and oceanic crusts which controls long-term continental stability.

## Results

### JHZs parental melt modelling

We derived the $SiO_2$ content and Si+O isotopic values of detrital zircon parental melts, by first modeling their parental melt trace element chemistry by combining in-situ zircon analyses (Supplementary Fig. 1) and new experimentally derived partition coefficients (Supplementary Datasets 1 and 2; See Methods for details). Then, using the melt Th/Y vs. $SiO_2$ calibration of ref.[6], we calculated the $SiO_2$ wt.% of JHZ parent

[1]Department of Earth & Environmental Sciences, University of Rochester, Rochester, NY 14627, USA. [2]School of Earth and Environmental Sciences, University of St Andrews, Bute Building, St Andrews, Scotland KY16 9TS, UK. ✉e-mail: wchowdhu@ur.rochester.edu

melts (Supplementary Dataset 2). The JHZ crystallization temperatures were calculated using the Ti-in-zircon thermometer[17]. This is a required constraint because of the derived melt temperature and SiO$_2$ wt.% both control zircon-melt Si and O isotope fractionation factors[18,19]. Finally, we combined fractionation factors with in-situ zircon Si and O isotopic measurements to derive the melt Si and O compositions.

Using this approach, we calculated the Si and O isotopic melt values for each individual Hadean and Eoarchaean zircon analysis (Supplementary Datasets 3 and 4), which we used as a petrological descriptor of early Earth crustal lithologies. Specifically, individual JHZ melt Si+O isotopic values can identify if supracrustal or altered sources −such as clays, shales, cherts, silicified pillow basalts, and/or serpentinites−could have been involved in the formation of the aforementioned melts (while O isotopes in serpentinites have been well studied, Si isotopes for the same are not, and so we also present new Si isotopic data from modern serpentinites (Supplementary Dataset 4)). Ultimately, each JHZ analysis provides constraints on a parental melt's SiO$_2$ wt.%, trace and rare-earth elements, and Si/O isotope compositions, which allows us to infer the possible lithological and tectonic character of these zircon-bearing Hadean rocks.

### JHZs parent melt silica content

Given that Th and Y are both high-field strength elements, the silicate melt Th/Y should be well correlated with the melt SiO$_2$ content[6]. We used this correlation to derive the melt silica content (see Methods) which is a constraint required to derive the Si/O isotopic chemistry the Eoarchean/Hadean melts. We first measured the concentration of Th and Y of JHZs and for these zircon data to be useful, we require zircon-melt partition coefficients (i.e., concentration in zircon divided by concentration in melt; $D^{zrc/melt}_{element}$) to derive the modeled Th/Y values of the JHZ melts. We conducted zircon-melt partitioning experiments (between 1300 and 975 °C) in a peraluminous and "wet" granitic system (Supplementary Dataset 5). We then calculated D-values for Th and Y at the crystallization temperatures of our JHZs, which were used to derive the melt Th and Y concentrations (See Methods and

Supplementary information; Supplementary Datasets 1 and 2). Depending on the crystallization T of our JHZs, the D-values for Th and Y were between 1.6−4 and 4.7−9.1, respectively. Our calculated melt Th and Y concentrations were combined with the empirical calibration of ref. [6] to estimate melt SiO$_2$ content (Fig. 1; Supplementary Dataset 2). The calculated SiO$_2$ contents range from 50 to 76 wt% (excluding one calculation at 81%) with an average value of 58 ± 9 wt% (2 s.d.), and ~95% of melts plot within the modern-day lower/upper continental crust field. Finally, we used the Ti-in-zircon crystallization thermometer[17] to determine zircon crystallization temperature (Supplementary Fig. 2). The crystallization temperatures range from 606 °C−803 °C with an average of 684 ± 78 °C (2 s.d.) (Supplementary Fig. 2a). Supplementary Fig. 2a indicates that most of our JHZs crystallized from intermediate to felsic melts with high water activity.

### JHZs parent melt Si and O isotopic content

To derive the δ$^{30}$Si and δ$^{18}$O melt values, we first performed in-situ coupled $^{30}$Si/$^{28}$Si and $^{18}$O/$^{16}$O ion microprobe measurements on our JHZs (Supplementary Fig. 1; Supplementary Dataset 3). Along with temperature, the primary melt property that affects mineral-melt fractionation factors is the melt silica content and the resulting polymerization of Si-O bonds[18]. Thus, the SiO$_2$ wt.% and crystallization T derived previously allows us to obtain zircon-melt fractionation factors for Si and O isotopes using experimental and empirical calibrations[18,19] (see Methods), and thereafter the Si and O isotopic composition of the melt (Fig. 2, Supplementary Dataset 4). These modelled melt data are far more implicative for deconvolving the petrogenesis of Hadean melts, as we can now compare these isotopic compositions to possible source lithologies (rather than just zircon data alone[20], see Supplementary Fig. 1).

## Discussion

With these modeled melt isotopic values, we can explore what their constituent lithologies might have been by comparing them to modern analogues. For comparison, we assume that the modern lithologies

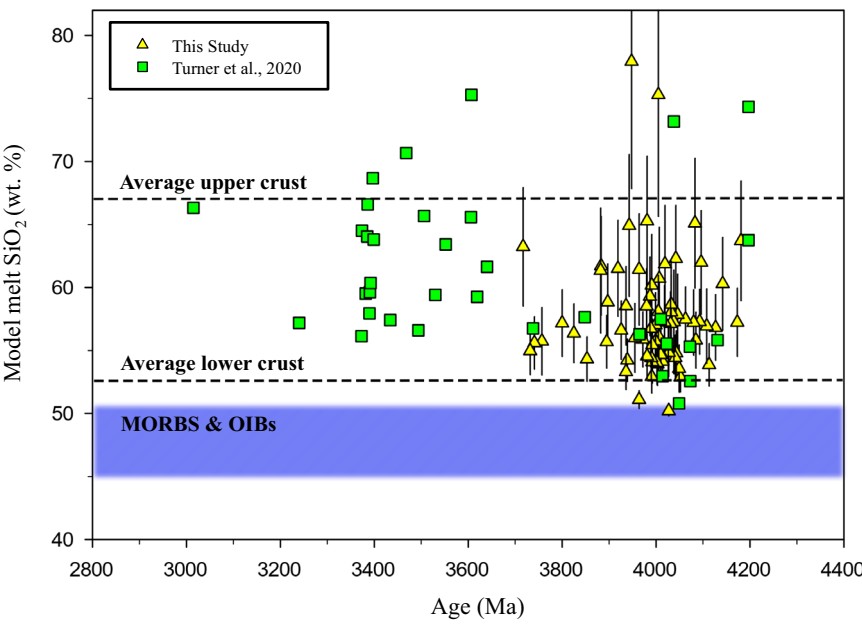

**Fig. 1 | Calculated SiO$_2$ wt% of modeled Jack Hills zircons' parent melts ($n$ = 77) against zircon $^{207}$Pb-$^{206}$Pb age.** The melt SiO$_2$ contents are calculated using new trace element partition coefficients reported here (58 ± 9 wt% (2 s.d.)) and are compared to those reported in ref. [6] (62 ± 17 wt% (2 s.d.)), using the same $D$ values. 18 of 95 JHZ measurements were rejected (see Methods for rejection criteria); also shown are results from ref. [6] ($n$ = 35). Field legend: horizontal dashed lines:

preferred upper and lower crust values define the upper and lower bound respectively. Blue field: All MORBs (Mid-Oceanic Ridge Basalts) and OIB (Ocean Island Basalt) values. The average upper and lower crust values are from ref. [34]. MORB and OIB values from ref. [32]. Error bars (2 s.e.) are propagated from the analytical errors of [Th] and [Y], and those of their $D$ values. (Details in Methods).

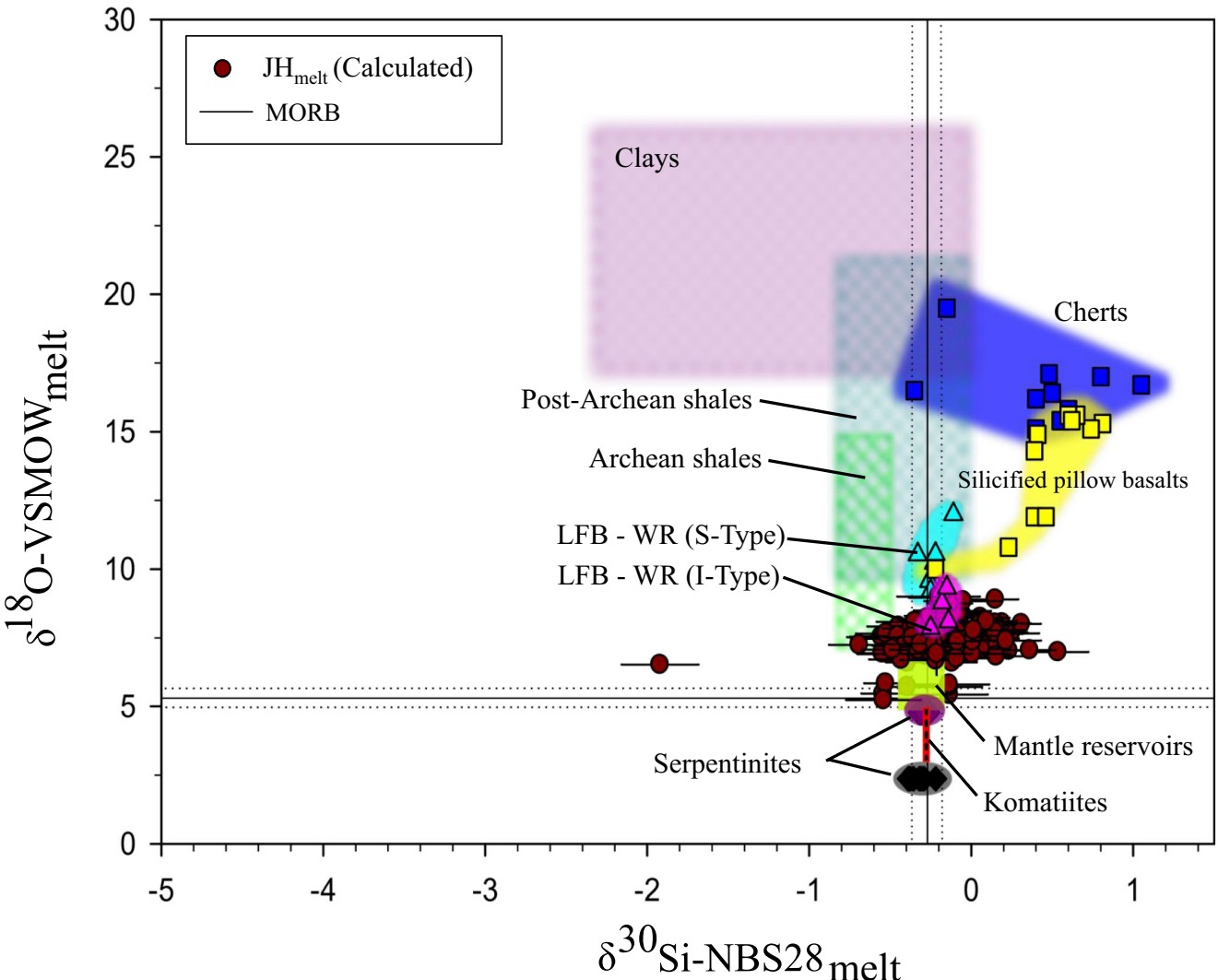

**Fig. 2 | $\delta^{30}Si_{NBS28}$ and $\delta^{18}O_{VSMOW}$ (±2 s.e; see Methods.) values of modeled JH (Jack Hills) melts ($n = 127$) in comparison with younger lithologies.** Individual analyses are presented here since some zircons could accommodate multiple ion microprobe sputtering locations. Sources: LFB WR (Lachlan Fold Belt Whole Rock; cyan and pink triangles and fields)[21], Archaean cherts/silicified pillow basalts from Barberton, South Africa (dark blue and yellow squares and fields)[49] and modern ODP (Ocean Drilling Program) serpentinites (black and maroon diamonds) ($\delta^{18}O$[50]). Clays (dark pink field) ($\delta^{30}Si$;[51] $\delta^{18}O$[52]), shales (thatched light green and turquoise fields) ($\delta^{30}Si$;[53] $\delta^{18}O$[54]), Mantle reservoirs (lime green box) ($\delta^{30}Si$ of HIMU (Hi-μ; Mangaia, Cook Islands), EM1 (Enriched Mantle 1; Pitcairn islands) and EM2 (Enriched Mantle 2; Samoa))[55]; $\delta^{18}O$ of HIMU[56] (Olivine phenocrysts[57], $\Delta^{18}O_{melt-olivine}$[58]), $\delta^{18}O$ of EM1 and EM2[56]) and Archaean komatiites (red bar) ($\delta^{30}Si$;[26] $\delta^{18}O$[59,60]). $JH_{melt}$ $\delta^{18}O$, serpentinite error bars are smaller than the symbols. The assimilation/re-melting of these lithologies was also hinted at in just the JHZ (Jack Hills zircon) isotopic data (Supplementary Fig. 1). Data for MORB (Mid-Oceanic Ridge Basalt) is from ref. [61].

shown in Fig. 2 are chemically analogous to similar rocks formed in the Archaean and Hadean.

We compared our calculated parent melt $\delta^{30}Si$ vs. $\delta^{18}O$ values to granitoids from the Lachlan Fold Belt (LFB)[21], supracrustal lithologies, end-member mantle reservoirs (MORBs, EM1 (Enriched Mantle 1), EM2 (Enriched Mantle 2) and HIMU (High-μ)), and Archaean komatiites to quantify possible interacting reservoirs and lithologies that could have created the JHZ melts. Our comparison to the primary mafic lithologies (mantle reservoirs, MORBs, and komatiites) shows that model melts are heavier in terms of their $\delta^{18}O$ values, and they have a larger $\delta^{30}Si$ range. Even if these primary melts were to assimilate serpentinites with a Si and O isotopic signature like ODP holes 1268 A and 1274 A (Supplementary Dataset 4), this composition alone cannot explain the derived Si/O melt values. Secondly, Si isotopic evolution of a melt by pure magmatic differentiation is broadly constrained to -0.3‰ for $SiO_2$ contents from ~45 to 75 wt%[21], and thus cannot create the range of our JHZ $\delta^{30}Si$ model melt values (Fig. 2). Thus, the assimilation of lithologies with more variable $\delta^{30}Si$ and elevated $\delta^{18}O$ needs to be invoked.

The lithologies with more variable $\delta^{30}Si$ and elevated $\delta^{18}O$ values (Fig. 2) are clays, shales, Archean cherts, and silicified pillow basalts. These lithologies represent common crustal lithologies and span a large range of $SiO_2$ content, consistent with the calculations presented in Fig. 1. Moreover, clays and shales are composite detritus that can be assumed to have been derived from an assortment of sub-aerial lithologies.

Inspecting Fig. 2 reveals that the mafic lithologies, clays, shales, Archean cherts, and silicified pillow basalts could have been re-melted to create the JHZ melts. The overlap between JHZ melts and the Phanerozoic granitoids implies that some early Earth melts share chemical similarities with modern I-type (igneous precursor) granitoids. However, when considering the isotopic values for Archaean shales, sediments like this could have contributed to the isotopic character of the generated melts, so Hadean S-type (sedimentary precursor) parent melts are also permissible.

If we consider the Si isotopic data in isolation, there is a clear overlap between the $\delta^{30}Si$ values of JHZ melts to recently reported

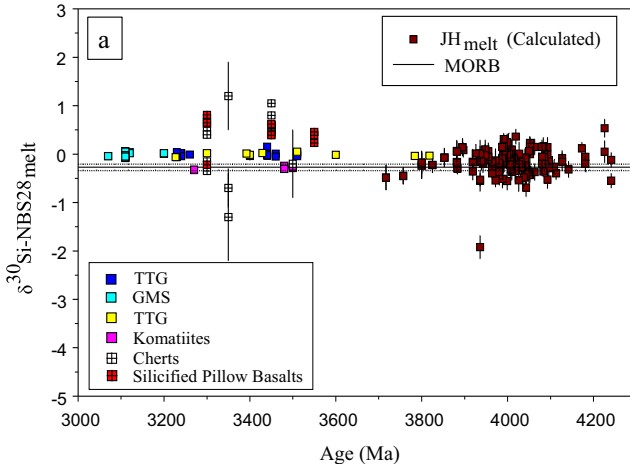
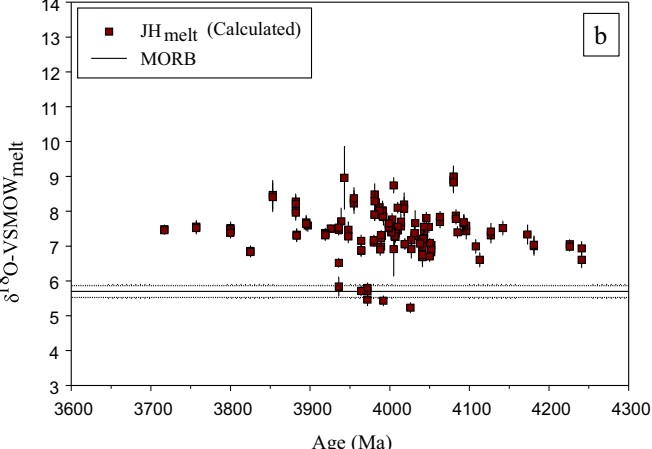

**Fig. 3 | JH (Jack Hills) model melt Si and O isotope content plotted separately. a** $\delta^{30}$Si of JHZ (Jack Hills Zircons) parent melts ($n = 127$) vs. age. Calculated JH melts are compared to Archaean rocks. **b** $\delta^{18}$O of JHZ parent melts ($n = 127$) vs. age. Data sources: Cherts[23,49]/Silicified pillow basalts[49]; TTGs[25,26] (Tonalite-Trondjhemite-Granodiorite); GMS[25] (Granite-Monzogranite-Syenite); Komatiite[26]; MORB[61] (Mid Oceanic Ridge Basalt). Error bars on our data are internal s.e.

values of Archaean cherts, silicified pillow basalts, TTGs, GMSs (Granite-Monzogranite-Syenite) and komatiites (Fig. 3a). The JHZ model melts also require lithologies with a large $\delta^{30}$Si range[22–24] that may be assimilated to generate the melt values we derived. Material analogous to Archaean chert and silicified pillow basalts (Fig. 3a; green/red crossed squares) are possible lithologies involved in shifting the $\delta^{30}$Si JHZ melt values to greater-than-MORB values given that, on balance, JHZ parent melts $\delta^{30}$Si values are slightly heavier ($-0.18 \pm 0.05$‰ (2 s.e.)) than the MORB reference ($\delta^{30}$Si $= -0.27$‰). To test this observation, we performed a t-test which compared the average JHZ model melt $\delta^{30}$Si value to the MORB reference and the test shows the average model melt $\delta^{30}$Si value is statistically different from the MORB reference (see Methods). Similar arguments have been made to justify the $\delta^{30}$Si values of Archean TTGs[25,26]. A recent study[27] proposed an empirical relationship between $\Delta^{30}$Si$_{melt-zircon}$ and SiO$_2$ wt.% of parent melt using natural zircons from various locations. Using this calibration[27], the average $\delta^{30}$Si values of our modeled JHZ melts increases from $-0.18$ to $-0.07$‰. This change does not influence our inferences of the JHZ model melts being isotopically heavier than MORB values or that cherts/silicified pillow basalts could have been assimilated to generate our JHZ model melts. In fact, the use of this new result would make our case even stronger as it shifts the calculated $\delta^{30}$Si of the melts to heavier values. Nevertheless, we have decided to use the calibration of[20] instead, because it is based on experiments conducted in a thermodynamically controlled environment, which is not subject to the complexities/uncertainties of a natural evolving igneous system.

The involvement of the source lithologies discussed previously imply the deposition of detrital sediments, precipitation of cherts, low/high-T water-rock interaction to form serpentinites and silicification of pillow basalts, their re-melting, and the evolution of the composite melt before the oldest zircon reported here crystallized (~4.2 Ga). Thus, a suitable tectonic regime needs to account for all these processes and the possible constituent lithologies. One possibility is a plate-boundary regime analogous to a modern subduction zone[6] where the above-mentioned lithologies are recycled and interact with mantle reservoirs. Subduction zone properties implied by JHZs are that the JH parent melts may have been intermediate to felsic, high in water activity, similar in its volatile content to modern arcs and formed in a low-T (650–800 °C), high-P (>4 kbar) regime[2,7,16,28]. As a test for this possible formational regime, we compare our modeled JHZ melt trace element ratios to modern lavas, which are presented on discriminatory diagrams that can fingerprint tectonic settings of rocks and minerals[29–34].

We compared key trace element ratios of JHZ model melts (calculated using our new partition coefficients) to the same trace element ratios of regimes where crust is generated on Earth (Fig. 4a, b). We also include trace element ratios of Lunar basalts and Martian meteorites fields as possible analogues for stagnant-lid tectonic scenarios with a caveat that these extra-terrestrial regimes might be anhydrous. The Th/Nb proxy (Fig. 4a) is useful in differentiating between oceanic basalts and lavas with a continental input. Thorium and Nb have similar compatibility during partial melting of the mantle to form basalts, but continental input increases the Th content of melts. The Dy/Yb ratio (Fig. 4b) is used as a proxy for calc-alkaline differentiation as well as a proxy for garnet. Silica content and Dy/Yb are negatively correlated while the crystallization of garnet decreases the Dy/Yb ratio in the restite melt[31]. Most of our melt values lie in the overlapping region between the island arc and continental arc lava field in Fig. 4a. In Fig. 4b, data generally fall in the island arc lava field, except for seven that plot in the overlapping region between the island arc and continental arc lava fields. Based on Fig. 4, we find that JHZs strongly suggest formation in a regime with subduction zone-like chemical characteristics. This observation and the dissimilarities with MOR and OIB lavas, lunar and martian rocks places specific chemical constraints on the JHZ formational environment and provides a limiting scenario about the geodynamics of the early Earth. Finally, a system as complex as a subduction regime that comprises several reservoirs such as fluxing fluids, mantle wedges, subducting slabs, sediments, etc. may require multiple avenues of inquiry when attempting to define its signature chemistry. Our study of TEs, combined with isotopic measurements, presents one such avenue. Additionally, we have also explored the Al content of our zircons which is the third most abundant element in the crust. The zircon Al content helps qualify the aluminosity of the parent melts[35], which can be a proxy for the incorporation of eroded subaerial crustal material. We find that a sizeable proportion (~18%) of the zircons indicate peraluminous (moles of Al$_2$O$_3$ > moles of Na$_2$O + K$_2$O + CaO) melts (Supplementary Fig. 2b).

The presence of peraluminous melts imply possible assimilation of metasedimentary material, partial melting of hydrous mafic parent material, and/or late stage differentiation of I-type granitic magmas[35]. This is similar to our observation made previously that S-type granitoids/clays/shales may be involved in generating JHZ parent melts.

To conclude, we propose that water-rock interactions took place at the Earth's surface to generate Hadean cherts, silicified pillow basalts, serpentinites, and detrital material. These lithologies were subsequently re-melted to generate felsic, peraluminous/

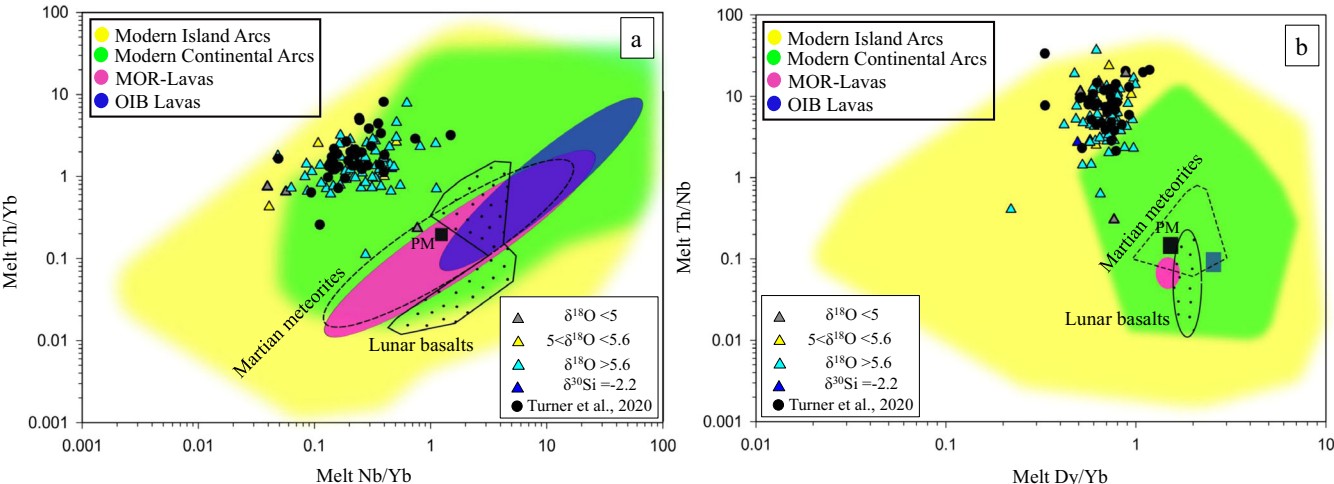

**Fig. 4 | Comparison of elemental ratios of derived melt values between our JHZs (Jack Hills Zircons; red, yellow, cyan, and blue triangles) and those reported in ref. [6] (black circles) to modern lavas. a** Th-Nb proxy used to classify lavas from different regimes, where lunar and Martian samples are shown as possible stagnant-lid analogues[62]. Field legend: OIB lavas (Ocean Island Basalt; Blue), MOR lavas (Mid Oceanic Ridge; Red), Modern arc lavas (Green/Yellow). Data sources: MOR + OIB Lavas[32]; Arc lavas compiled from the GEOROC database (Island arcs: Aleutian arc, Honshu arc, Tonga, Mariana arc. Continental Arcs: Andean arc, Cascades, Alaskan peninsula); Lunar basalts[62] and Martian meteorites[62] **b** Dy/Yb proxy for calc-alkaline metaluminous melts, based on Th/Y, and further supported by zircon differentiation as well as for restite garnet along with the Th/Nb proxy. Data sources: MOR + OIB lavas[29] Arc lavas from GEOROC; Lunar basalts[63], Martian meteorites[64]. Primitive Mantle (PM; Black square)[65]. Trace element melt contents were derived from zircon using newly derived partition coefficients (Supplementary Fig. 5). Errors for our data are given in Supplementary Dataset 2. A modified version of this figure (Supplementary Fig. 8) is in the Supplementary information file that contains TTG (Tonalite-Trondjhemite-Granodiorite) fields (Data from GEOROC database).

metaluminous melts, based on Th/Y, and further supported by zircon Al contents that constrain melt aluminosity (Supplementary Fig. 2b, c). Moreover, these interactions and re-melting processes gave all our JHZ model melts Si and O isotopic composition that do not vary with time and thus these processes might have been uniformly operating from 4.2 to 3.7 Ga. Finally, whatever the precise details are of the regime that did create these zircons, our calculated melt compositions show clear chemical similarities with modern plate boundaries. This period (500–700 Myr) of our planet is especially important now as we begin to explore the habitability of other terrestrial-like planets some of which may be like the Hadean Earth.

## Methods

### U-Pb geochronology

Zircons were analyzed for their U-Pb age with the University of Rochester Cetac (formerly Photon Machines) Analyte G2 193 nm excimer Laser Ablation (LA) system coupled to an Agilent 7900 quadrupole ICP-MS (Inductively Coupled Plasma Mass Spectrometer). The LA system has a HelEx sample chamber that uses a He as a carrier gas. The He flow rates were set at 0.6 L/min in the sample chamber and in the carrier gas tube upon exiting the chamber at 0.2 L/min for a total flow rate of 0.8 L/min. The laser energy set point was 7 mJ, with a spot size of 25 μm which results in a fluence of 11.81 J/cm². The laser beam was pulsed at 10 Hz. Each ablation included a collection of background counts for 20 s, followed by an ablation period of 20 s and finally, a washout period of at least ~30 s prior to the next analysis. We analyzed $^{91}$Zr, $^{202}$Hg, $^{204}$Pb, $^{206}$Pb, $^{207}$Pb, $^{208}$Pb, $^{232}$Th, and $^{238}$U. The integration time for $^{206}$Pb and $^{207}$Pb was 30 ms and the for the rest of the isotopes, it was 10 ms for each cycle.

An initial population of ~2000 zircon grains were HF treated and firstly mounted on double-sided sticky tape, without embedding them in epoxy or polishing. These grains were screened for their $^{207}$Pb/$^{206}$Pb ratio (Supplementary Dataset 6), and zircons with a $^{207}$Pb/$^{206}$Pb corresponding to a date of 3.6 Ga or older were hand-picked and mounted along with a Si and O isotopic standard (KIM) on double-sided tape. The grains and standards were cast in epoxy within the inner 1 cm diameter of a 2.55 cm (1″) diameter mount. The sample was then polished by hand to expose the cores of the grains. These older grains were then analyzed for their Si and O isotopes (details in the next section). Afterwards, grains were analyzed for their U-Pb age again where the subsequent LA-ICP-MS spot is placed directly on top of the stable isotope data acquired by ion microprobe. These data were collected using the same LA-ICP-MS, and this U-Pb age is reported in Supplementary Dataset 7. We used the natural AS3 zircon[36] as an age standard and the Iolite 3.32 software package[37] to derive the final age values. We also used a Kuehl Lake zircon (chemically similar to the 91500 zircon) as a secondary age standard). After the second U-Pb age measurements, all the grains had a degree of concordance between 87 and 105%

$$\text{Concordance}\,(\%) = \left[\frac{206/238\ age}{207/206\ age} * 100\right] \tag{1}$$

and were selected for discussion in the manuscript; most of our concordance values (~70%) are between 98 and 102%. Past stable isotope studies of JHZs that are as much as 15% discordant are representative of primary stable isotope data[38].

### Si and O isotopes

For the stable isotope measurements, we used the CAMECA-ims1290 Secondary Ion Mass Spectrometer (SIMS) housed at the University of California, Los Angeles. We used a 3-nA Cs+ primary beam, rastering over 20 × 20 μm on the samples, that yielded secondary ion signals (O$^{18-}$ and Si$^{30-}$ ≥6 × 10⁶ and 3 × 10⁶ counts per second, respectively) that were collected with Faraday cups (FCs) in dynamic multicollection mode. This configuration allows for simultaneous measurement of $^{16}$O$^-$ and $^{18}$O$^-$ on the L′2 and H′2 FCs, respectively, followed by that of $^{28}$Si$^-$ and $^{30}$Si$^-$ on C and H1 (all FCs) after only one mass jump. We used a mass resolution (M/ΔM) of 2,400 to separate molecular interferences from peaks of interest. Each spot analysis comprised 14 cycles, each of which includes a counting time of 10 s for oxygen isotopes, and 15 s for Si isotopes. The backgrounds of FCs were determined during the 90 s pre-sputtering before each analysis. In some cases, the JH zircons were large enough to contain multiple SIMS spots and so individual analyses

are reported rather than individual zircons. The zircons were standard-sample bracketed; we measured approximately three standard analyses after every five unknown analyses. The stable isotope standard KIM[20], a mantle-derived zircon from Kimberley Pool, South Africa, was used as a drift-correcting standard and to derive the δ- values for both isotopic systems. For the δ[30]Si derivation, each unknown was calibrated against the average [30]Si/[28]Si of KIM analyses immediately preceding and succeeding the unknown.

After our SIMS analyses, the zircons were imaged using an SEM at UCLA and the grains with visible cracks in the SIMS sputtering pit were flagged and not considered in the discussion. The JH Si and O isotopic values are reported as $\delta^{30}Si_{NBS28}$ ‰ and $\delta^{18}O_{VSMOW}$ ‰ where

$$\delta^{30}Si = \left( \frac{(^{30}Si/^{28}Si)_{sample}}{(^{30}Si/^{28}Si)_{standard}} - 1 \right) * 1000 \qquad (2)$$

and

$$\delta^{18}O = \left( \frac{(^{18}O/^{16}O)_{sample}}{(^{18}O/^{16}O)_{standard}} - 1 \right) * 1000 \qquad (3)$$

The derived Si and O δ-values were adjusted to account for the δ[30]Si and δ[18]O offset between KIM and NBS28 (−0.38‰) as well as between KIM and VSMOW (5.35‰)[20].

To better understand the internal zoning characteristics of the zircons—e.g., to interrogate whether a particular grain or grain region comprises what is characteristically interpreted to be igneous zoning—we performed cathodoluminescence imaging on the zircons (CL) (Supplementary Fig. 3). These images also serve as a guide to better understand the intragrain context of the acquired Si and O isotopes results. The CL images were taken using a Cameca SXFive Electron Microprobe housed at Syracuse University.

We have also calculated the Si and O isotopic content of purported parent melts of the JH zircons by using $\Delta^{30}Si_{melt-zircon}$ and $\Delta^{18}O_{melt-zircon}$ values obtained from two sources[18,19] and using the crystallization temperatures and the derived melt SiO₂ wt.% for the zircons reported here. We use a published relationship[18] between $\Delta^{30}Si_{melt-zircon}$, SiO₂ wt% of parent melt and the crystallization T based on whole rock and zircon measurements on tonalites from Western Australia while[19] proposes an empirical relationship between melt SiO₂ wt.% and $\Delta^{18}O_{melt-zircon}$ derived from samples from the Sierra Nevada batholith.

### Si/O isotopic composition of JHZs

Supplementary Fig. 1 shows the Si/O isotopic values for our JHZs. Figures like Supplementary Fig. 1 have been used previously[21,26,27] to infer constituent lithologies, though calculations for the melt Si and O isotopic composition are possible now with characterized zircon-melt fractionation factors for Si and O, as well as a means to quantify the melt SiO₂ content. The bulk of our sample set of Hadean and Eoarchean zircons lie above the range of δ[18]O mantle zircon values (Supplementary Fig. 1). There are also JHZs with higher-than mantle and lower-than mantle δ[30]Si. There is also a minor population of analyses that seem to possess mantle values. Finally, there is one analysis which is a δ[30]Si outlier at −2.2‰. This analysis also has a mantle value in terms of δ[18]O. This low value may be because of the incorporation of cherts and silica precipitates (Fig. 3a), especially those associated with BIFs[39]. However, three analyses were conducted on this grain and this outlying value is not reproduced by the two other analyses.

### ODP serpentinites

To assess the degree of Si isotope fractionation that is associated with the bulk-rock serpentinisation of mafic and ultramafic rocks, a suite of variably serpentinised abyssal peridotites have been analysed for their whole-rock Si isotope compositions. The samples were taken from cores drilled on the Ocean Drilling Program Leg 209 cruise[40], whose main aim was to drill mantle peridotites and associated mafic rocks from fracture zones along the Mid-Atlantic Ridge[41] The samples were taken from two cores, Holes 1274 A and 1268 A. Briefly, Hole 1274 A is composed of harburgites, dunites, and gabbroic rocks, all of which had undergone 60–100% serpentinisation at low temperatures (100–200 °C). Major element analyses suggests that the serpentinisation was isochemical, i.e., the rocks from Hole 1274 A were hydrated, but no other metasomatism took place. Hole 1268 A also returned a mixture of harzburgites, dunites and gabbros; in contrast to Hole 1274 A, the rocks from 1268 A were 100% serpentinised before undergoing a late-stage, high temperature (>200 °C) metasomatic event (associated with a gabbroic intrusion) which resulted in the variable desilicification of some rocks to form talc (steatisation).

To specifically study the potential effect of serpentisation (both isochemical and metasomatic) on the Si isotope composition of ultramafic rocks, only variably altered, serpentinised harzburgites were chosen from both cores for analysis. Six samples were taken from Hole 1274 A and eleven samples were taken from Hole 1268 A. Samples were provided in powder form, and prepared for Si isotope analysis via solution MC-ICP-MS following the methods first described in[42] and further detailed in[43].

Sample powders were dissolved using alkali fusion, whereby ~10 mg of sample powder was weighed into silver crucibles along with ~200 mg of NaOH flux (semiconductor grade, Merck). The crucibles were placed into a muffle furnace heated to 720 °C for 15 min to perform the fusion. Subsequently, the crucibles and fusion cake were placed into individual PFA vials filled with 20 ml of MQ-e water and left overnight to equilibrate. The fusion cake was then transferred via pipette to precleaned PP bottles, diluted with enough MQ-e to reach a final Si concentration of between 10 and 25ppm, and acidified with enough conc. HNO₃ to bring the pH of the solution to ~2. Final Si concentrations of the sample solutions were ascertained via the Heteropoly Blue method using a photospectrometer.

Samples were purified for Si isotopes using a single stage cation exchange procedure. Samples were loaded into BioRad Polyprep columns filled with 1.8 ml of AG50W X12 cation exchange resin (200–400 mesh, BioRad; the resin was cleaned in the column before samples were loaded—see[43] for the cleaning method). Silicon is in either anionic or neutral forms in solution at low pH and so can be directly eluted using 5 ml of MQ-e water. Post-column, the samples were acidified to ~0.22 M HNO₃. Total procedural blanks of the whole chemical preparation procedure were measured at less than 100 ng of Si, which is approximately 0.35% of the total measured Si signal and considered negligible.

Silicon isotopes were measured on a Neptune Plus MC-ICM-MS instrument (ThermoFischer Scientific, Bremen, Germany), running at medium resolution (M/ΔM ~7500) to avoid significant molecular interferences on the ²⁹Si and ³⁰Si beams. Samples were introduced into the instrument using a 100 µl min-1 PFA ESI (Elemental Scientific, Omaha, USA) microflow nebulizer running into the SIS spray chamber. Silicon isotopes were measured in the L3 (²⁸Si), C (²⁹Si), and H3 (³⁰Si) Faraday cups, and depending on instrumental conditions, a 2ppm Si beams typically gave a total analyte signal of ~7 V. Isotope ratios were measured in static mode with each measurement consisting of 25 cycles with a ~8 sec integration time.

Silicon isotope measurements were calculated using the standard-sample bracketing protocol relative to the NBS28 standard. Variations in Si isotopes are defined using the delta notation (δ[30]Si $_{NBS28}$) as described before. Each measurement session consisted of 10 samples, two of which were always the external standards BHVO-2 and Diatomite. Long-term δ[30]Si error on the NBS28 bracketing standard over the analytical sessions is 0.00 ± 0.10‰ (2 s.d.).

The δ[18]O values used in this study are those from different serpentinite samples but of the same ODP holes measured by ref. [43].

## TE and REE measurements in zircon

We used the same LA-ICP-MS instrument to analyze for TE and REE as we did for the U-Pb geochronology. The instrumental parameters were mostly the same except for the fluence, ablation period, and washout time which were at 6.72 J/cm$^2$, 20 s, and 35 s respectively. The isotopes analyzed were $^{27}$Al, $^{29}$Si, $^{31}$P, $^{45}$Sc, $^{49}$Ti, $^{58}$Fe, $^{88}$Sr, $^{89}$Y, $^{91}$Zr, $^{93}$Nb, $^{181}$Ta, $^{232}$Th, $^{238}$U, $^{139}$La, $^{142}$Ce, $^{141}$Pr, $^{147}$Sm, $^{153}$Eu, $^{157}$Gd, $^{164}$Dy and $^{172}$Yb. We took care to select areas as close to the igneous cores of each of the zircons of interest which we identified from CL images. We also plotted the REE values of our zircons (Supplementary Fig. 4) to help establish their igneous character. We calibrated the elemental raw data for all elements (apart from Ti) using NIST 610 as the primary standard and NIST 612 and Kuehl Lake (KL) zircon (similar to 91,500[44]) as a secondary standard. For Ti, we used KL as the primary standard. Iron-58 (analyses >1 ppm were rejected), $^{139}$La(analyses >5 ppm were rejected), and Th/U (analyses with 0.1>Th/U > 1 were rejected) were used as markers to reject analyses that might contain significant contributions possibly from inclusions. One analysis was rejected that had [Al] = 420 ppm which is much higher than the values reported by ref. [35] for JHZs. Titanium-49 was used to derive the crystallization temperature of the zircons using the calibration of[17] and assuming unity activities for SiO$_2$ and TiO$_2$. We used $^{93}$Nb, $^{232}$Th, $^{164}$Dy, and $^{172}$Yb to create discrimination diagrams to determine the JH formational environment. Melt [Th]/[Y] was used to derive the SiO$_2$ content of the parent melts of the JHZs using the calibration mentioned in ref. [6]. The melt values of Th, Y, Nb, Dy, and Yb were derived using the experimentally derived partition coefficients (see next section for the experimental methods) and used in discrimination diagrams. They were also compared to JH melt data derived from zircon data reported by ref. [6].

## TE/REE partitioning experiments and measurements

To augment the zircon-melt partition coefficients for trace and rare-earth elements reported by[45] who derived the same for anhydrous experiments conducted at 1300 °C and 1 atm in alkali-free melts, we conducted new zircon-melt experiments (Supplementary Dataset 1) using a piston cylinder device (Supplementary Dataset 1 and Supplementary Fig. 5). The D-values of[45] were used by[6] along with those of[46], to characterize JH parent melts.

We measured zircon-melt partition coefficients of Sr, Nb, Th, Y, Dy, and Yb, at five Ts between 1300 °C and 975 °C and 1 GPa, where the silicate melt was peraluminous and H$_2$O-bearing. The temperature was chosen to be high enough to produce synthetic zircon crystals that are large enough to be measured by LA-ICP-MS. The experiments were designed with sufficient differences when compared to the study of[45]; our experimental mix was made more alkali and water-rich. We wanted to evaluate whether temperature, pressure, water, and alkali-content of the melt could yield different D values for the selected elements.

The starting mix for our experiments and the experimental details are given in Supplementary Dataset 5. This mix was prepared by first mixing alkali carbonates and ZrO$_2$. Then we decarbonated the mix by holding it at 850 °C for 5 h in a muffle furnace. This mix was then held at 1350 °C in a muffle furnace overnight and then dropped into water to be quenched into glass. Following the quenching step, we mixed the glass with Al(OH)$_3$ and SiO$_2$ in an agate mortar to make a peraluminous granitic mix. Separately, we made a trace element mix by first decarbonating SrCO$_3$ at 1200 °C for 2 h in a muffle furnace to make SrO, which we then added to a mix containing the rest of the trace element as oxides. The final experimental mix was created by dry mixing the trace element mix and the peraluminous granitic mix. For all our experiments, we loaded this final mix into a 3 mm diameter platinum capsule along with 18.2 MΩ·cm water to bring the total final water content to ~10%. We then inserted the Pt capsule into an Mo sleeve and covered it with a Pt foil, which we then capped with a 2 mm thick Mo lid. We fit this entire assembly into an MgO sleeve and inserted it into a graphite furnace with MgO filler pieces at the top and bottom. This setup was fit into a borosilicate sleeve. We then placed three hollow cylinders made of compressed NaCl around the borosilicate, wrapped it in Pb foil, and inserted the assembly into a piston cylinder pressure vessel. We followed the same mix-creating procedure as established in[7]. The experiments were conducted by first pressurizing the assembly to 1 GPa, displayed on a 20 cm Heise pressure gauge, and heated up to the temperature of interest at 100 °C/min. The temperature was monitored using a W$_{75}$Re$_{25}$ and W$_{97}$Re$_3$ thermocouple wire procured from Concept Alloys Inc. (with an accuracy of ±3 °C) encased in a mullite (Al$_2$O$_3$) thermocouple sheath. Once the experiment had run its course (48–336 h.), the melt in the capsule was quenched by shutting off power to the instrument.

After the completion of each experiment, the capsule was sliced in half and mounted on a 1" epoxy round and polished using 1 μm alumina powder and colloidal silica in preparation for LA-ICP-MS measurements. We analyzed the experimental zircon and glass with the Cetac (formerly Photon Machines) Analyte G2 193 nm excimer LA system coupled to an Agilent 7900 quadrupole ICP-MS (Inductively Coupled Plasma Mass Spectrometer) housed at the University of Rochester. Helium was used as a carrier gas to transport ablated material to the ICP-MS. Helium flow rates in the HelEx sample chamber was at 0.6 L/min and in the carrier gas tube upon exiting the chamber at 0.2 L/min. Before entering the ICPMS we also introduced Ar at 1.3 L/min as an additional carrier gas.

We analyzed the experimental zircons by pulse-firing the laser at a fluence of 4.06 J/cm$^2$ for 15 s with a pulse frequency of 10 Hz and a spot size of 5 μm. Before and after the ablation time were a 20 s period during which a background signal was collected and a 35 s washout time before the next analysis. We followed the same analytical procedures employed by[7] to analyze experimental zircons and glasses. We collected $^{23}$Na, $^{27}$Al, $^{29}$Si, $^{88}$Sr, $^{89}$Y, $^{91}$Zr, $^{93}$Nb, $^{232}$Th, $^{164}$Dy and $^{172}$Yb for the zircons and added $^{27}$Al, $^{39}$K and $^{48}$Ca for the glasses.

## Results and extrapolated D values

The calibrations for all the elements that we have used in the main manuscript are shown in Supplementary Fig. 5. Our D values are either similar (Y, Yb, and Dy) to or lower than (Th, Nb, and Sr) all the previously reported D values at 1300 °C of[45] and[47]. That said, our experimental mix is far more applicable to a natural intermediate/felsic system. However, if we were to report the extrapolated D-values at the T of[46] and[48], and compare them to the reported values of these two studies, our extrapolated values are much lower. We ascribe the differences between our experimental data and the natural data of[46] and[48] to the differences in SiO$_2$% of the two systems under consideration[46]; and[48] investigate natural systems that are highly silicic (~75% SiO$_2$) compared to our experimental charge (61–63% SiO$_2$).

For comparison, we have also regressed our dataset after combining them with the natural datasets of[46] and[48]. The resultant D values from this different regression and the subsequent discrimination diagrams are shown in Supplementary Fig. 6 and 7. While Supplementary Fig. 7 is different from Fig. 4, the modeled JHZ melt values still lie in the arc lava fields and thus the inferences presented in the main section of the manuscript remain unchanged.

## Statistical tests

**$t$ test comparisons between Hadean and Eoarchean melt $\delta^{30}$Si values, and the MORB reference ($\delta^{30}$Si = −0.27‰)**

H$_0$ (Null hypothesis): $\mu_{JHZ\ melt}$ = −0.27

H$_a$ (Alternate hypothesis): $\mu_{JHZ\ melt}$ ≠ −0.27

Where, $\mu_{JHZ\ melt}$ = mean $\delta^{30}$Si value

For JHZ melts,

| | $\delta^{30}Si$- |
|---|---|
| Mean ($\mu_H$) | −0.18 |
| Std. error | 0.026 |

$\alpha = 0.05$
Hypothesized mean = −0.27
*t*-statistic = 10.4
P ($t = 10.4$) = $1.57 \times 10^{-18}$
Since $P < \alpha$, $H_a$ is accepted[6]

**Calibration equations used in the study**
**To derive SiO₂ %:**

$$\left(\frac{Th}{Y}\right)_{melt} = 0.0269 \times (SiO_2)_{melt} - 1.3169 (R^2 = 0.9812) \quad (4)$$

This equation was derived by regressing whole-rock data from over 40,000 analyses for rocks between 50 and 70% SiO₂[6]. The calibration is unfiltered for age. Refrain from using an Archean-only dataset, but rather prefer a dataset populated with more temporal and compositional diversity.

**To derive Δ³⁰Si:**

$$1000 ln\alpha^{30/28Si}_{melt-zrc} = A \times \frac{10^6}{T^2(K)} \quad (5)$$

This equation was obtained from[18]. Where,

$$A = 0.25 \times SiO_2(\%) - 0.01 \quad (6)$$

Data used to derive (6):

| From Trail et al., 2019 | |
|---|---|
| SiO₂ (%) | A |
| 100 | 0.53 |
| 65 | 0.39 |
| 50 | 0.08 |

**To derive Δ¹⁸O[19]:**

$$1000 ln\alpha^{18/16O}_{WR-zrc} = 0.0612 \times SiO_2(\%) - 2.5 \quad (7)$$

**Error calculations in the manuscript**

$$\sigma_{Silica\%} = \sigma_{meltTh/Y} / 0.0269 \quad (8)$$

$$\text{Where,} \ \sigma_{meltTh/Y} = \sqrt{\left(\frac{D_Y}{D_{Th}} * \sigma_{\frac{Th}{Y}Zrc}\right)^2 + \left(\frac{Th}{Y}_{Zrc} * \sigma_{\frac{D_Y}{D_{Th}}}\right)^2} \quad (9)$$

$$\text{Where,} \ \sigma_{D_Y/D_{Th}} = \frac{D_Y}{D_{Th}} \times \sqrt{\left(\frac{\sigma_{D_Y}}{D_Y}\right)^2 + \left(\frac{\sigma_{D_{Th}}}{D_{Th}}\right)^2} \quad (10)$$

$$\sigma_{\delta^{30}Si_{melt}} = \sqrt{\left(\sigma_{\Delta^{30}Si}\right)^2 + \left(\sigma_{\delta^{30}Si_{Zrc}}\right)^2} \quad (11)$$

$$\text{Where,} \ \sigma_{\Delta^{30}Si} = \left(\frac{10^6}{T^2(K)}\right) \times \sqrt{\sigma_A^2 + \left(\frac{2A\sigma_T}{T(K)}\right)^2} \quad (12)$$

$$\text{Where,} \ \sigma_A = \sigma_{SiO_2(\%)} \times 0.0081 \quad (13)$$

$$\sigma_{\delta^{18}O_{melt}} = \sqrt{\left(0.0612 \times \sigma_{SiO_2(\%)}\right)^2 + \left(\sigma_{\delta^{18}O_{Zrc}}\right)^2} \quad (14)$$

## Data availability
The authors declare that the data supporting the findings of this study are available within the paper and its supplementary information files.

## Code availability
This manuscript uses Iolite 3.32 to process raw LA-ICPMS data. The software has been published in a peer-reviewed journal[37]. The software is not freeware but a free trial version may be downloaded here https://iolite.xyz/.

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

## Acknowledgements

We would like to thank Dr. Ming-Chang Liu for their help in collecting Si and O isotope data, and Jacob Buettner for his support while collecting U-Pb age data. This work was supported by NSF EAR-1650033 (DT), NSF EAR-1751903 (DT) and NASA PC3 grant 80NSSC19M0069 (DT).

## Author contributions

The study was conceived by D.T. and P.S.S. M.M., D.T., and W.C. participated in the stable isotope measurements and data reduction. LA-ICP-MS measurements and data reduction was completed by W.C. with assistance from M.M. W.C. conducted the high T/P experiments and took measurements. The serpentinite analyses were done P.S.S. W.C. wrote the manuscript with significant input from D.T. and P.S.S.

## Competing interests

The authors declare no competing interests.
