## [Peer Review File · Nature Communications]

Eoarchean and Hadean melts reveal arc-like trace element and isotopic signaturesReviewer #1 (Remarks to the Author):

This manuscript by Chowhury et al. uses the composition of melts calculated to be in equilibrium with Jack Hills zircons to place constraints on the regime in which they formed and thus comment on Archaean geodynamics. The problem is a good one and has received a lot of attention so it is a suitable topic for Nature Communications.

The approach is relatively new but some of it has been done before and was published in the same journal. Namely, using partition coefficients and trace element concentrations to invert equilibrium melt compositions including using Th/Y ratios to derive melt SiO₂. Even the use of Th/Yb - Nb/Yb and Dy/Yb discrimination diagrams is identical. Whilst it is important to see the verification of the earlier work in a follow-up study a Nature Communications paper needs to be clear about what is new. This should be stated explicitly early on: this study verifies earlier results using the same approach BUT the new and novel aspect is (1) the new and arguably more appropriate partition coefficients and (2) the extrapolation to melt Si and O isotope ratios. The latter, has permitted, for the first time, identification of specific source components (chert, serpentinite) that had not been possible in the earlier work and there and further and independent evidence for subduction (i.e. reworked material that interaction with water were transported into the melting region. With that made clear the paper will be much stronger.

With regard (1) there should be a comment on why the new experiments might be more appropriate and since it is only DTh that differs from the Burnham and Berry work please comment on why this is different.

In lines 45 and 172 there is mention of the importance of Al but I could not find any discussion of this in the paper. It should be added if it currently is hidden in supplementary material or these references deleted.

In summary, I believe the paper will make a good contribution to Nature Communications but the authors have been unnecessarily timid about what has been done before (but now verified in this independent study) which distracts a bit from these very exciting new aspects of the Si and O isotopes.

Reviewer #2 (Remarks to the Author):

Review for Nat Comm
June 2022

General comments

The manuscript, "Eoarchean/Hadean melts reveal arc-like trace element and isotopic chemistry" argues that Hadean and Eoarchean detrital zircons from the Jack Hills region, Western Australia, record evidence for incorporation of a variety of surficial materials—shales, cherts, serpentinites, and altered pillow basalts—in their source magmas. This implies that surficial material was either taken to depth or assimilated in the source region. The authors take this further to suggest that these detrital zircons are sourced from arc-like melts and that the geodynamic regime under which they formed did not change from 4.2 to 3.6 Ga.

Overall, I find this manuscript to be unsuitable for publication in Nature Communications or anywhere else. This work is original, but the scientific quality is generally quite poor. Major components of the data validation and presentation are missing, there are logical flaws, and the paper is not well written. I was frustrated by this manuscript because the Jack Hills detrital zircons are a rare and precious resource of the earliest vestiges of Earth history, and I think that this poorly planned and poorly executed study was a waste of this resource.

I am providing my most substantial comments below.

Summary of most major suggestions for revisions

I. Overall framing

It seems like there is a somewhat strange bias in this paper by the subdivision of the detrital zircons into two groups—Hadean and “Archean” (which is really Eoarchean). This sets up an assumption/presupposition that something changed at the Hadean-Archean boundary, which isn’t something that has been suggested by other studies. The overall thesis is the continuity of magmatic processes from 4.2 to 3.6 Ga, which can be discussed separately from this Hadean-Archean dichotomy.

It was also unclear to me what conclusion(s) is/are novel in this manuscript in follow up to Trail et al (2018) PNAS, which is strangely not cited here?

Trail, D., Boehnke, P., Savage, P. S., Liu, M. C., Miller, M. L., & Bindeman, I. (2018). Origin and significance of Si and O isotope heterogeneities in Phanerozoic, Archean, and Hadean zircon. *Proceedings of the National Academy of Sciences*, 115(41), 10287-10292.

II. What does variation in $d_{30}\text{Si}$ really tell you?

Because the $d_{30}\text{Si}$ of melts and zircons that form from them depends on $[\text{SiO}_2]$ and thus the progression of fractional crystallization (e.g. Guitreau et al 2022 GCA, which is not cited here), it is unclear to me why the authors expect that the Si isotope composition should be expected to be a passive tracer of the incorporation of surficial materials into these magmas. I therefore do not think that the mixing model presented in Figure 2 is in the least bit convincing. I don’t understand why the authors are not speculating on how fractional crystallization and then subsequent remelting of protocrust or precursor source material could result in the spread of Si isotope compositions that they observe?

There are some fundamental questions about the application of Si isotopes that are not answered in this manuscript—e.g. is the range measured in the Jack Hills zircons reasonable for magmatic fractionation? Has anyone looked at multiple zircons in one magma, and how much can they vary? Is it possible to alter the $d_{30}\text{Si}$ in a radiation-damaged domain (which are pervasive in the Jack Hills zircons)?

Si is a major element. If it is partly sourced from sediments, as is interpreted in this manuscript, then these melts must have A LOT of sediment. Can the authors make an estimate of how much of this sedimentary material would need to be incorporated to result in the melt compositions they calculate?

III. Silicate melt Th/Y, Th/Nb, etc

In line 68 it says that ‘details and caveats’ for the application of the calibrated zircon Th/Y vs melt SiO_2 relationship are discussed in the Online methods, but they aren’t really. The relationship also isn’t plotted or really vetted in the original publication they cite (Turner et al). Why should we trust this? What is the sensitivity of it? Has the Turner publication done a sufficient job of determining the uncertainties? What about variable Th/Y within one zircon—is this something that you see in your dataset, and if so, how do you explain it?

Why are the calculated Melt Th/Nb so high relative to modern lavas? Is this a problem? It isn’t discussed and there should at minimum be speculation. This seems like misattribution of partition coefficients to me, if it doesn't match anything modern?

Why are the Jack Hills zircons, which have low crystallization temperatures consistent with formation in a late-stage siliceous melt, being compared with MORBs and OIBs as potential 'sources'? There is little primary zircon in MORBs and OIBs. Why not compare with TTGs or experimentally remelted MORB?

About 17% of Zr is ^{92}Zr , so $^{92}\text{ZrH}^+$ interference on the ^{93}Nb mass that was measured ought to be significant considering how much Zr there is in a zircon ablation. Therefore Nb in zircon is usually only measured by very high-mass resolution techniques—namely SHRIMP. For analysis on a quadrupole, how are they correcting or minimizing this interference? My suspicion is that this hasn't been accounted for and the Nb in zircon is likely overmeasured due to Zr interference so I would expect the melt Nb/Yb to be overestimated. I can't verify if this is the case because there isn't any reference material validation provided for the TE measurements (see last section).

IV. Temperatures of zircon formation and selection of partition coefficients

The partition coefficient experiments (comparison in ST1) show that partition coefficients are highly sensitive to temperature, yet the experiments from this study were performed at much higher temperatures (1300 and 1100C) than the Ti-in-zircon temperatures that are calculated in this study (~800-600C). It appears that this was done because the resultant zircons are sufficiently large for LA-ICP-MS characterization, but that isn't really an acceptable explanation—the authors clearly have accessed a SIMS facility, so why don't they analyze the TEs in smaller zircons formed at more appropriate temperatures for their partition coefficient determinations?

V. Imprecise language

Much of the language in this manuscript is imprecise and I would suggest going through it carefully so that it will be more digestible; as is, it is in places inaccessible and in others technically incorrect. I am including a few examples here; most of them are minor but the point I would like to get across is that this manuscript needs substantial revision to be up to the caliber of Nature Communications.

Major issues with grammar and flow:

Line 108: "The mafic lithologies, sediments and cherts/silicified pillow basalts seem to suggest that these might be possible rocks that could have re-melted to create the composite melts from which the JHZs crystallized."

Line 112: Low $\delta^{18}\text{O}$ relative to what? (precision of wording issue)

Trite and loose connection to life. Poorly written and difficult to follow:

Line 176: "We are proposing naturally and experimentally derived models that directly quantify the chemical and physical state of our planet during its evolution during the first 500 to 700 Myr when life might have emerged. This period of our planet is important now; since we are exploring the habitability of other terrestrial-like planets some of which may be like the Hadean Earth."

Line 34 wording is vague/misleading here and implies that was mobile-lid during Hadean, which is not the interpretation of the Aarons et al paper, which instead suggested subduction-like setting at 3.75 Ga.

These clauses do not belong together (i.e. the 'thus' does not make sense)

Line 42: "Additionally, we also explore the Al content of our zircons which is the third most abundant element in the crust and thus gives us insight into the aluminosity of the parent melts."

VI. Treatment of data

Overall, the presentation and treatment of data in this manuscript is much too sparse. Key information is missing. For the U-Th-Pb isotope data, only the final 207/206 date and its analytical uncertainty are presented. The full U-Th-Pb isotope dataset and all of the measured dates should be included in the table. Additionally, the reference material results are not presented. It seems that the authors have only used one reference zircon in their analysis, but the U-Pb community requires at least one secondary reference material as validation, as well, and that appears to be missing from the method. The age uncertainties are missing from all figures, yet they are important for considering temporal trends and therefore need to be included.

In general, the presentation of U-Pb isotope and TE data in this manuscript does not follow the typical reporting protocols of the zircon U-Pb and TE community and I would suggest presenting a more systematic summary of the secondary zircon reference material results for validation purposes. This should include propagation of within-session and long-term excess variances as appropriate. There also needs to be more detail regarding discordance cut-offs, common-Pb correction, etc. I would also include summary figures of zircon secondary refmat TE profiles with 2SD values for each TE so that the general degree of uncertainty of TE data is more apparent and is not just the internal 2SE from the measurement for each unknown. This will make the new data in this publication more citable in future studies.

I was also unsure about the quality of the oxygen isotope data. How did the authors account for radiation damage? Did they look at the hydration state of the zircon (OH/O measurement during O isotope analysis) to vet the quality of their results? Also, there is one calculated melt composition with an uncertainty of +/-1.1 permil, which is quite large—why is that?

Line 126 'maybe an inclusion' isn't taking it far enough—can you see this in the time-resolved LA data?

There should be CL images of every single zircon analyzed, with all analytical spots spatially indicated for reference. There are only 5 of these images and they do not include the analytical spots!

Reviewer #3 (Remarks to the Author):

In "Eoarchean/Hadean melts reveal arc-like trace element and isotopic chemistry," Chowdhury and colleagues use zircon from the Jack Hills of Western Australia to expand our understanding of Earth's earliest crust-forming processes. Demystifying this earliest chapter of Earth's history is important for constraining the onset of mobile lid tectonics, the establishment of permanent crust, and the initiation of nutrient cycling between land and ocean. It is also important for knowing how far back we can responsibly apply the logic of "the present is the key to the past". Investigating these earliest years of Earth's not-yet-rock record is challenging because we rely on evidence in the mineral zircon to reconstruct and interpret major earth cycles and processes. Chowdhury and colleagues constrain the TE/REE, SiO₂, and Si-O isotopic compositions of the parent melts from which Jack Hills zircons crystallized using new experimentally derived TE/REE partition coefficients, a Th/Y proxy for SiO₂, and isotopic fractionation factors. Using these calculated compositions as converging lines of evidence, the authors determine that zircons throughout the Hadean and early Archean likely crystallized from magmas generated in an arc-like setting. Evidence for mobile lid tectonics so early in Earth's history, and the continuity of setting and process throughout the Hadean and into the Archean, are both important contributions to the ongoing debate about the formation of Earth's earliest stable crust.

I'm grateful to have had the opportunity to review this manuscript. It was thought provoking and I learned a lot. As the authors say in their introduction, studies of the Hadean are "fraught with controversy," and I read the manuscript through that lens. Below I offer several comments and suggestions. Some of my suggestions may sound big (I think more discussion about, and evaluation of, the new partition coefficients is needed!), but they're all offered in the spirit of "accept with minor revisions." I hope that the authors' response to some or all of my comments will help make the manuscript even more compelling to a critical reader of Nature Communications. I'm certainly happy to take another look at a manuscript before it goes to press, should that be useful for the authors and/or the editor.

General comment:

Try to use "calculated melt" or "model melt" consistently throughout (instead of just "melt") when referring to compositions determined using partition coefficients or proxies or fractionation factors.

Line-by-line comments:

Line 19: Is it possible to distinguish between continental arcs and oceanic arcs using this dataset? (line 141 suggests you're thinking about island arcs)

Line 45: I think a reference to Ackerson et al. (2021) would be appropriate and useful here.

Line 45: I kept my eyes open for this discussion of Al-in-zircon because I'm pretty interested in it right now, but I didn't find it again in the main text until line 172, in the concluding paragraph. Figures S2A and S2B are interesting and thought provoking, but given the prominent mention of Al in the intro and conclusion, I'm surprised there's not more discussion of Al-in-zircon (and peraluminous/metaluminous melts) in the main text. If there's not enough space in the main text to add a few more words about this, please refer explicitly to Figure S2 in the text (if journal rules permit it) and expand the caption.

Line 50: "To derive the SiO₂ content and Si+O isotopic values of our zircons' parental melts, we first model their parental melt trace element chemistry

Line 53 / Line 68 / Online Methods Line 197: Is this discussion of GEOROC describing what was done by Turner et al. (2020), or new work you have done to test or refine the Th/Y proxy? If reporting previous work, I think the parenthetical explanation at line 53 can be cut for clarity (let the reference do the explaining). If new, please provide the search criteria and access date used to assemble the whole-rock dataset from GEOROC. I'd also be curious to know the logic behind using whole-rock data instead of volcanic glass and melt inclusions, which might be more relevant for constraining melt composition at the time of zircon crystallization. If there is no difference between Th/Y vs SiO₂ in whole rock compositions and in glass compositions, that would be interesting and important to know, too.

Line 59: "zircon analyzed" or zircon analysis? Did any crystals have multiple analytical spots?

Line 62: the parenthetical discussion of serpentinites is a little distracting here—can it be moved to a relevant figure caption, or elsewhere?

Line 73: I appreciate the importance of using partition coefficients tailored to your specific system. This is something I've thought a lot about, and I'm glad others are thinking about it, too! However, it's not clear to me why/how your D values are an improvement over other published D values. Why are they better suited for the JH zircons than other published partition coefficients out there. Should we all be using these new D values when we calculate model melts for the JH? If so, why? Tell us clearly! I like that you "wanted to evaluate whether temperature, pressure, water, and alkali-content of the melt could yield different D values" (online methods line 138-140), but it's not clear to me that you varied any parameters other than temperature (1100 vs 1300 C). The significance of the different D values at different temperatures would be useful to discuss, as would the similarities to the Burnham and Berry (2012) Ds, and the differences with the Padilla and Gualda (2016) Ds. It's possible I lost the thread as I moved back and forth between the main text, the online methods, supplemental table 1, supplemental table 5, and supplemental figure 5 trying to piece this together. Having a more centralized/streamlined presentation, evaluation, and discussion of these new partition coefficients would be very useful for readers like me who are excited by this work and want to follow the logic of it (in the supplement is fine). Claiborne et al. (2018) and Burnham (2020) have interesting discussions about environmental factors (e.g., temperature) influencing the partitioning of REE in zircon. It might be worthwhile to cite one or both of these papers in a future discussion.

Line 75: The D values determined at 1100 C? Why did you choose these instead of the ones determined at 1300 C? (I can guess, but your evaluation would carry more weight)

Line 76: I'm very curious, would your model melt SiO₂ compositions have been substantially different (enough to alter your final interpretations) if you had calculated Th/Y ratios determined using different partition coefficients? (the higher T ones you calculated, plus the B+B and P+G ones used in your Fig S5, maybe also Sano et al., 2002 and Claiborne et al., 2018)? Interesting if the answer is yes, interesting if the answer is no. It would be great to know!

Line 107: Reference 21 (Savage et al., 2011)—is this paper (largely focused on Hekla, Iceland) the one you meant to cite here?

Line 119-120 / Figure 3A: Clarify—is the d_{30Si} mantle value the same as the MORB reference value?

Line 141: Carley et al. (2022) came to a similar conclusion, but reported greater similarities with modern continental arcs than modern island arcs.

Line 142 + 145: include the word "calculated" or "model" to describe the JHZ melt

Line 146: Clarify—Turner et al. (2020) used different D and also different zircon crystals and data?

Line 153-155: I think a reference to Davidson et al. 2007 and/or 2013 would be appropriate and useful

Line 161-164: continue to emphasize "calculated" melt.

Line 161-164: "one" and "a couple" and "the majority" are a little unsatisfying without being reminded of how many calculated melts you have in total. Anchor your reader with some real numbers.

Figure comments:

Figure 1: Explain in the caption, what is the significance of 720C? Why is this the dividing line for differently colored symbols? Why not (for example) the average temperature of 685 that you told us about in the text?

Figure 1: Consider adding a dashed line (or similar) to show the Hadean-Archean boundary (important to clearly establish the age you use for the next figures).

Figure 1: It would be interesting to add Iceland in addition to MORB and OIB (same comment applies to Figure 4)

Figure 1: please add n-values to caption or legend

Figure 1 caption: The difference in model melt SiO₂ is interesting (and exciting!), but I need to know, why are your D values better suited to this effort than the ones presented by Turner et al. (2020) who used Burnham and Berry (2012)? I'm ready to be persuaded!

Figures 1 and 4: I'm curious—if you used Turner et al. (2020) data and your D values, or your data with the Burnham and Berry (2012) D values used by Turner et al. (2020), would trends change substantially? If this is what you've already done, please clarify!

Figure 2: This is a very data-rich figure! It's overwhelming to untangle what is in the legend, what is labeled directly, and what is named/described in the caption. There seems to be a lot going on in the olive green box, and it's hard to spot. The two circular symbols below the horizontal MORB line are unclear/confusing--are these fields for the serpentinites? Please try to streamline (maybe more labels? Maybe more in the legend?)

Figure 3A: Is the chert reference in the legend intentionally? Seems out of place. `

Figure 3A: Make clearer in the legend (not just the main text at line 118) the difference between the green-box chert and the gray-box chert.

Figure 3 (A+B): Add n-values to legend or caption

REVIEWER COMMENTS AND RESPONSES

Reviewer #1 (Remarks to the Author):

This manuscript by Chowhury et al. uses the composition of melts calculated to be in equilibrium with Jack Hills zircons to place constraints on the regime in which they formed and thus comment on Archaean geodynamics. The problem is a good one and has received a lot of attention so it is a suitable topic for Nature Communications.

The approach is relatively new but some of it has been done before and was published in the same journal. Namely, using partition coefficients and trace element concentrations to invert equilibrium melt compositions including using Th/Y ratios to derive melt SiO₂. Even the use of Th/Yb - Nb/Yb and Dy/Yb discrimination diagrams is identical. Whilst it is important to see the verification of the earlier work in a follow-up study a Nature Communications paper needs to be clear about what is new. This should be stated explicitly early on: this study verifies earlier results using the same approach BUT the new and novel aspect is (1) the new and arguably more appropriate partition coefficients and (2) the extrapolation to melt Si and O isotope ratios. The latter, has permitted, for the first time, identification of specific source components (chert, serpentinite) that had not been possible in the earlier work and there and further and independent evidence for subduction (i.e. reworked material that interaction with water were transported into the melting region. With that made clear the paper will be much stronger.

With regard (1) there should be a comment on why the new experiments might be more appropriate and since it is only D_{Th} that differs from the Burnham and Berry work please comment on why this is different.

Based partially on this comment and those made by another reviewer regarding partition coefficients, we decided to expand upon the partition coefficient (D-values) by performing three more experiments so that we may extrapolate and derive the partition coefficients for all elements involved at the T of crystallization for the zircons. We agree with the reviewer that the arguments for propriety of the partition coefficients used, needs to be more robust. This re-evaluation of the partition coefficients is the most significant change from the original manuscript, but it does improve the science of the same. Thanks for this comment.

In lines 45 and 172 there is mention of the importance of Al but I could not find any discussion of this in the paper. It should be added if it currently is hidden in supplementary material or these references deleted.

Al measurements have been discussed in the main section and relevant figures have been presented in the Extended Data Figures file.

“...The presence of peraluminous melts imply late stage differentiation of I-type granitic magmas, possible assimilation of metasedimentary material or partial melting of hydrous mafic parent material at

>7 kbar³⁵. This is similar to our observation made previously that S-type granitoids/clays/shales may be involved in generating JHZ parent melts.”

In summary, I believe the paper will make a good contribution to Nature Communications but the authors have been unnecessarily timid about what has been done before (but now verified in this independent study) which distracts a bit from their very exciting new aspects of the Si and O isotopes.

Reviewer #2 (Remarks to the Author):

Review for Nat Comm

June 2022

General comments

The manuscript, "Eoarchean/Hadean melts reveal arc-like trace element and isotopic chemistry" argues that Hadean and Eoarchean detrital zircons from the Jack Hills region, Western Australia, record evidence for incorporation of a variety of surficial materials—shales, cherts, serpentinites, and altered pillow basalts—in their source magmas. This implies that surficial material was either taken to depth or assimilated in the source region. The authors take this further to suggest that these detrital zircons are sourced from arc-like melts and that the geodynamic regime under which they formed did not change from 4.2 to 3.6 Ga.

Overall, I find this manuscript to be unsuitable for publication in Nature Communications or anywhere else. This work is original, but the scientific quality is generally quite poor. Major components of the data validation and presentation are missing, there are logical flaws, and the paper is not well written. I was frustrated by this manuscript because the Jack Hills detrital zircons are a rare and precious resource of the earliest vestiges of Earth history, and I think that this poorly planned and poorly executed study was a waste of this resource.

It is saddening to hear such a radically negative summary of our hard work. We are aware of how precious the Jack Hills zircons are, and hope to have done them justice. As the reviewer might expect, we cannot agree with the statement that our science is “generally quite poor” – we have utilised established analytical techniques and we believe that we have couched our study within the framework of the state-of-the-art science. We hope that our comments, and the revised manuscript changes their mind.

I am providing my most substantial comments below.

Summary of most major suggestions for revisions

I. Overall framing

It seems like there is a somewhat strange bias in this paper by the subdivision of the detrital zircons into two groups—Hadean and “Archean” (which is really Eoarchean). This sets up an assumption/presupposition that something changed at the Hadean-Archean boundary, which isn’t something that has been suggested by other studies. The overall thesis is the continuity of magmatic processes from 4.2 to 3.6 Ga, which can be discussed separately from this Hadean-Archean dichotomy.

The division at 4 Ga has been used merely in keeping with the division proposed by the ICS. Because this division invokes the presupposition as described by the reviewer, we have decided to combine both zircon populations into one.

It was also unclear to me what conclusion(s) is/are novel in this manuscript in follow up to Trail et al (2018) PNAS, which is strangely not cited here?

Trail, D., Boehnke, P., Savage, P. S., Liu, M. C., Miller, M. L., & Bindeman, I. (2018). Origin and significance of Si and O isotope heterogeneities in Phanerozoic, Archean, and Hadean zircon. *Proceedings of the National Academy of Sciences*, 115(41), 10287-10292.

In this revised manuscript, we have experimentally calibrated the change in D-values of the trace/rare-earth elements, in a hydrous silicate melt, versus $10^4/T$ (K) for all elements discussed here. This allows us to derive the D-value at the crystallization T of the JHZs. As far as we know, this treatment of D-values has not been done before. However, this is only one facet of the work presented. These results are fed into models to compare JHZ equilibrium melt composition to more modern lavas to infer geodynamic environments more robustly. Secondly, this is also the first time that the isotopic values of JHZs have been used to derive $d^{18}\text{O}$ and $d^{30}\text{Si}$ Archean and Hadean melt composition by combining multiple experimental and empirical calibrations. There were three pieces of information missing when Trail et al. (2018) paper was published. First, there was no published experimental $d^{30}\text{Si}$ fractionation data between zircon and other igneous phases (rectified by Trail et al. 2019). Second, the information in Turner et al. (2020), which presents a way to “map” zircon trace element data to SiO_2 melt content was not published in 2018. Third there were no datasets that enabled Si and O zircon isotope data to be combined trace element data to estimate SiO_2 content AND the Si and O isotopic composition of the parental melt. By putting all these pieces together, this gives us a direct insight, for the first time, into what the chemistry of Eoarchean/Hadean melts were, which is pertinent since we do not have adequate samples from this time. Moreover, as noted above, we provide zircon-melt partition coefficients derived from melts that are H_2O and alkali bearing (cf Burnham and Berry 2012).

The suggested reference has been added.

II. What does variation in $\delta^{30}\text{Si}$ really tell you?

Because the $\delta^{30}\text{Si}$ of melts and zircons that form from them depends on $[\text{SiO}_2]$ and thus the progression of fractional crystallization (e.g. Guitreau et al 2022 GCA, which is not cited here), it is unclear to me why the authors expect that the Si isotope composition should be expected to be a passive tracer of the incorporation of surficial materials into these magmas. I therefore do not think that the mixing model presented in Figure 2 is in the least bit convincing. I don't understand why the authors are not speculating on how fractional crystallization and then subsequent remelting of protocrust or precursor source material could result in the spread of Si isotope compositions that they observe?

The Guitreau et al. 2022 GCA paper was published after we submitted our manuscript but we have discussed their results in the revised version.

Let us consider the reviewer's suggestion: **"...fractional crystallization and then subsequent remelting of protocrust or precursor source material could result in the spread of Si isotope compositions"**

The range of $\delta^{30}\text{Si}$ for zircons from I-type magmas is around -1 to -0.1‰ (Guitreau et al., 2022). Assuming an average $\Delta^{30}\text{Si}$ of 0.2 (This study), the range of $\delta^{30}\text{Si}$ for modeled I-type magmas is -0.8 to 0.1‰. Any resultant melt generated from any combination of these magmas would be some unique value within this range. This value does not reflect the range of values we observe for our JHZ model melts (-0.9 to 0.7‰). Thus, we need to consider the remelting of supracrustal and detrital material.

Secondly, We have plotted modern S/I-type granites and all possible mafic end-members along with Komatiites as well as cherts with SiO_2 close to 100%. Thus, we have considered primary and detrital lithologies with a wide range of SiO_2 contents.

We hope this makes Figure 2 more convincing.

There are some fundamental questions about the application of Si isotopes that are not answered in this manuscript—e.g. is the range measured in the Jack Hills zircons reasonable for magmatic fractionation? Has anyone looked at multiple zircons in one magma, and how much can they vary? Is it possible to alter the $\delta^{30}\text{Si}$ in a radiation-damaged domain (which are pervasive in the Jack Hills zircons)?

As we mention above, because it is limited at magmatic temperatures, we do not think magmatic differentiation alone can account for our observed $\delta^{30}\text{Si}$ values nor do we claim as such as explained above. Guitreau et al., 2022 reported the $\Delta^{30}\text{Si}$ between zircon and carbonatitic melt (15 wt. % SiO_2) and compared it to the experimental $\Delta^{30}\text{Si}$ between zircon and quartz ($\text{SiO}_2 = 100\%$) from Trail et al., 2019. The entire range is 0.5‰ which does not explain the whole range of JHZ Si isotope compositions.

Guitreau et al., 2022 also reported $\delta^{30}\text{Si}$ of zircons from I-type granites. The greatest 2 s.d. that they report, of all the zircons from I-type granites in their study, is 0.42‰ which is much lower than our observed range.

Our zircons were HF treated and radiation damaged regions/grains are unlikely to be analyzed and reported. In case they do make it through the HF filter, we CL imaged our zircons and targeted regions that are typically attributed to igneous zoning (keeping in mind that pattern recognition is imprecise,

and unlikely to yield identical interpretation from one practitioner to the other). We have also rejected analyses that show cracks in the sputtering locations. We have provided CL images of the entire suite of accepted zircon analyses along with sputtering locations.

Si is a major element. If it is partly sourced from sediments, as is interpreted in this manuscript, then these melts must have A LOT of sediment. Can the authors make an estimate of how much of this sedimentary material would need to be incorporated to result in the melt compositions they calculate?

The exact proportions of each possible precursor may be theoretically derived through modelling and simulations but that is a separate exercise beyond the scope of this manuscript. For some context, Trail et al, 2018 present a simple mixing model between a basalt and chert and they report that an 80%-20% = basalt-chert mix can account for the $\delta^{30}\text{Si}$ of natural zircon that they report. Moreover, given the proximity of the d18O values of our modeled JHZ melts to primary igneous material, a “lot” of sediments are not really required. Given the similarity in $\delta^{18}\text{O}$ and $\delta^{30}\text{Si}$ between our JHZ model melts and modern LFB granites (Fig. 2), we would speculate that similar amounts of non-igneous material were involved in generating the JHZ melts as were involved in generating the LFB granitoids. This is the advantage of using coupled Si+O isotopic values; if we were only looking at $\delta^{30}\text{Si}$ values, then the reviewer’s comment is difficult to rebut, but when considering the constraints placed by the d18O values, we are on safe ground making this inference.

III. Silicate melt Th/Y, Th/Nb, etc

In line 68 it says that ‘details and caveats’ for the application of the calibrated zircon Th/Y vs melt SiO₂ relationship are discussed in the Online methods, but they aren’t really. The relationship also isn’t plotted or really vetted in the original publication they cite (Turner et al). Why should we trust this? What is the sensitivity of it? Has the Turner publication done a sufficient job of determining the uncertainties? What about variable Th/Y within one zircon—is this something that you see in your dataset, and if so, how do you explain it?

One of the objectives of this manuscript is to derive JHZ melts’ SiO₂ content. The Turner et al 2020 paper provides us with a novel and valid way of estimating it using zircon chemistry; if the reviewer has an issue with this calibration, then it is not our place to argue with them – none of us on this paper feature in the Turner et al author list; as it is in a peer-reviewed journal, and having read and agreed with the article’s approach, we choose to give it credence as an acceptable method.

To reiterate, the Turner et al (2020) calibration uses the non-controversial observation that because Th and Y are HFSEs, they are both melt-incompatible elements, and should be well-correlated with SiO₂%. The large dataset and the low uncertainties of the Turner et al calibration (Supplementary Figure 1 in Turner et al., 2020) provides even more confidence to this method.

Perhaps the biggest uncertainty for the calibration is the D-values for both Th and Y at JHZ crystallization temperatures; therefore, in responding to this criticism, we have performed new experiments to better constrain these values. Thorium and Y D-values ($[\text{Th}]_{\text{zrc}}/[\text{Th}]_{\text{melt}}$, etc.) lie between 1.5 and 5, and 4 and 10 respectively, which agree broadly with previous work. Our new data add further credence and

strengthens the Turner et al. calibration in that they relied on an experimental partitioning study with a melt composition that is demonstrably non-natural.

We did not measure each zircon more than once and thus we do not know about intra-crystal variations. However, zircons can only form in felsic/late-stage siliceous melts and thus even if a zircon were to show intra-crystalline variation in Th/Y, they should not have a large variation in SiO₂%. Ignoring one exception (81%) the rest of the zircons have melt SiO₂% values between 50-76% corresponding to an average melt Th/Y ratio of 0.23 ± 0.03 ($2 \times$ s.e.). This is similar to the value for the bulk continental crust (0.3; Rudnick and Gao, 2003) and we do not expect intracrystalline variations to be different from this range.

Why are the calculated Melt Th/Nb so high relative to modern lavas? Is this a problem? It isn't discussed and there should at minimum be speculation. This seems like misattribution of partition coefficients to me, if it doesn't match anything modern?

We have re-hauled the partition coefficients and discrimination diagrams presented in this manuscript. We do so because we discovered that data used by Turner et al., 2020 was not comprehensive. In the new diagrams, we have included a larger arc lava dataset from the GEOROC database where we have separate fields for island arcs and continental arcs. Our entire JHZ model melt dataset now lies within the arc lava fields and thus strengthens our inference that our JHZ model melts display arc-like chemistry.

Why are the Jack Hills zircons, which have low crystallization temperatures consistent with formation in a late-stage siliceous melt, being compared with MORBs and OIBs as potential 'sources'? There is little primary zircon in MORBs and OIBs. Why not compare with TTGs or experimentally remelted MORB?

Significant zircons can be found in gabbros, and while their REE patterns may resemble continental crust zircons, HFSEs have been used to discriminate between zircons from oceanic crust and those from the continental crust. Grimes et al., 2007 have found that zircons from these two sources have distinct HFSE signatures even though they formed from late-stage siliceous melts. Early felsic crust may have formed from hydrated basaltic crust (Drabon et al., 2021, Borisova et al., 2022) and that is why we have included these mafic rock types in our comparison. As mentioned by Harrison et al., 2017, if JHZs did crystallize from TTGs, they only did so from a late-stage siliceous melts when the parent melt was probably not a TTG and thus TTGs have not been included. Also, TTGs have Sr/Y values between 20-500 (Hoffman et al.,

2019 while our JHZ model melts have Sr/Y ranging between 2 and 11 (excluding two model melts at 21 and 31). In case the reviewer is interested, here is Fig. 4 with an added TTG field (Data from GEOROC database).

About 17% of Zr is ^{92}Zr , so $^{92}\text{ZrH}^+$ interference on the ^{93}Nb mass that was measured ought to be significant considering how much Zr there is in a zircon ablation. Therefore Nb in zircon is usually only measured by very high-mass resolution techniques—namely SHRIMP. For analysis on a quadrupole, how are they correcting or minimizing this interference? My suspicion is that this hasn't been accounted for and the Nb in zircon is likely overmeasured due to Zr interference so I would expect the melt Nb/Yb to be overestimated. I can't verify if this is the case because there isn't any reference material validation provided for the TE measurements (see last section).

We do acknowledge that we should have provided standard verification of TE values for readers. With this in mind, we provide measurements of Keuhl Lake zircon (**[Nb] = 0.81 ± 0.04 ppm**), chemically similar to 91500 zircon, as a secondary standard and have found them to be similar to reported TE concentrations for 91500 (**[Nb] = 0.79 ppm**). We have reported these values in this revised version in the supplementary tables. The Nb concentrations in our experimental zircons is between 10-30 ppm. Interference from $^{92}\text{ZrH}^+$, if as significant as the reviewer suggests, should have resulted in much higher Nb concentrations. Moreover, if $^{92}\text{ZrH}^+$ had in fact swamped Nb measurements, we would not have seen any trend in log D-values versus $10^4/T$.

IV. Temperatures of zircon formation and selection of partition coefficients

The partition coefficient experiments (comparison in ST1) show that partition coefficients are highly sensitive to temperature, yet the experiments from this study were performed at much higher temperatures (1300 and 1100C) than the Ti-in-zircon temperatures that are calculated in this study (~800-600C). It appears that this was done because the resultant zircons are sufficiently large for LA-ICP-MS characterization, but that isn't really an acceptable explanation—the authors clearly have accessed a SIMS facility, so why don't they analyze the TEs in smaller zircons formed at more appropriate temperatures for their partition coefficient determinations?

A SIMS may not provide improved spatial resolution and so we have not used it for our small experimental products. The zircons grown at 975 °C were barely measurable using a 4 μm spot size and this experiment was run for 336h. For some context, the JHZs grew to about 30-40 μm at ~680 °C over perhaps 10^6 years. So growing experimental zircons at more appropriate temperatures might take inordinately long.

As explained previously, we conducted experiments to be able to derive more appropriate D-values.

V. Imprecise language

Much of the language in this manuscript is imprecise and I would suggest going through it carefully so that it will be more digestible; as is, it is in places inaccessible and in others technically incorrect. I am including a few examples here; most of them are minor but the point I would like to get across is that this manuscript needs substantial revision to be up to the caliber of Nature Communications.

Major issues with grammar and flow:

Line 108: "The mafic lithologies, sediments and cherts/silicified pillow basalts seem to suggest that these might be possible rocks that could have re-melted to create the composite melts from which the JHZs crystallized."

It has been fixed.

Line 112: Low d18O relative to what? (precision of wording issue)

It has been fixed.

Trite and loose connection to life. Poorly written and difficult to follow:

Line 176: "We are proposing naturally and experimentally derived models that directly quantify the chemical and physical state of our planet during its evolution during the first 500 to 700 Myr when life might have emerged. This period of our planet is important now; since we are exploring the habitability of other terrestrial-like planets some of which may be like the Hadean Earth."

The words "when life might have emerged" has been removed. We believe the rest is relevant to the study.

Line 34 wording is vague/misleading here and implies that was mobile-lid during Hadean, which is not the interpretation of the Aarons et al paper, which instead suggested subduction-like setting at 3.75 Ga.

The reference has been removed.

These clauses do not belong together (i.e. the 'thus' does not make sense)

Line 42: "Additionally, we also explore the Al content of our zircons which is the third most abundant element in the crust and thus gives us insight into the aluminosity of the parent melts."

This has been changed to "Additionally, we also explore the Al content of our zircons which is the third most abundant element in the crust. The Al content helps us in qualifying the aluminosity of the parent melts."

VI. Treatment of data

Overall, the presentation and treatment of data in this manuscript is much too sparse. Key information is missing. For the U-Th-Pb isotope data, only the final 207/206 date and its analytical uncertainty are presented. The full U-Th-Pb isotope dataset and all of the measured dates should be included in the

table. Additionally, the reference material results are not presented. It seems that the authors have only used one reference zircon in their analysis, but the U-Pb community requires at least one secondary reference material as validation, as well, and that appears to be missing from the method. The age uncertainties are missing from all figures, yet they are important for considering temporal trends and therefore need to be included.

For the geochronology two standards (Keuhl lake and AS3) were used. Keuhl Lake is a secondary laboratory standard believed to be from the same locality as 91500 and used multiple times by us for geochronology, trace element and stable isotope verification purposes (Trail et al. 2015; Trail et al. 2017; Trail et al. 2018; Chowdhury et al. 2020). The full raw data table with errors and all isotopes has been provided in this manuscript.

In general, the presentation of U-Pb isotope and TE data in this manuscript does not follow the typical reporting protocols of the zircon U-Pb and TE community and I would suggest presenting a more systematic summary of the secondary zircon reference material results for validation purposes. This should include propagation of within-session and long-term excess variances as appropriate. There also needs to be more detail regarding discordance cut-offs, common-Pb correction, etc. I would also include summary figures of zircon secondary refmat TE profiles with 2SD values for each TE so that the general degree of uncertainty of TE data is more apparent and is not just the internal 2SE from the measurement for each unknown. This will make the new data in this publication more citable in future studies.

We have provided the raw geochronology data for the first exploratory LA-ICPMS measurements as well as the second geochronology exercise that gave us ages that we have used in the manuscript.

Zircons TE data for the secondary zircon standard has also been provided.

I was also unsure about the quality of the oxygen isotope data. How did the authors account for radiation damage? Did they look at the hydration state of the zircon (OH/O measurement during O isotope analysis) to vet the quality of their results? Also, there is one calculated melt composition with an uncertainty of +/-1.1 permil, which is quite large—why is that?

The zircons were HF treated which should be a good filter to exclude radiation damaged regions/grains. Moreover, the O isotope data are virtually identical to other studies that have presented stable O isotope data in JH zircons for about the 15 years.

We are not sure where the reviewer sees the data point with the high uncertainty – this datum does not exist in our study.

Line 126 ‘maybe an inclusion’ isn’t taking it far enough—can you see this in the time-resolved LA data?

We have reported the location of this analysis on a CL image. The location does seem to have sampled a non-igneous region of the grain.

There should be CL images of every single zircon analyzed, with all analytical spots spatially indicated for reference. There are only 5 of these images and they do not include the analytical spots!

This has been done. The full suite of zircons have been provided.

Reviewer #3 (Remarks to the Author):

In “Eoarchean/Hadean melts reveal arc-like trace element and isotopic chemistry,” Chowdhury and colleagues use zircon from the Jack Hills of Western Australia to expand our understanding of Earth’s earliest crust-forming processes. Demystifying this earliest chapter of Earth’s history is important for constraining the onset of mobile lid tectonics, the establishment of permanent crust, and the initiation of nutrient cycling between and land and ocean. It is also important for knowing how far back we can

responsibly apply the logic of “the present is the key to the past”. Investigating these earliest years of Earth’s not-yet-rock record is challenging because we rely on evidence in the mineral zircon to reconstruct and interpret major earth cycles and processes. Chowdhury and colleagues constrain the TE/REE, SiO₂, and Si-O isotopic compositions of the parent melts from which Jack Hills zircons crystallized using new experimentally derived TE/REE partition coefficients,

a Th/Y proxy for SiO₂, and isotopic fractionation factors. Using these calculated compositions as converging lines of evidence, the authors determine that zircons throughout the Hadean and early Archean likely crystallized from magmas generated in an arc-like setting. Evidence for mobile lid tectonics so early in Earth’s history, and the continuity of setting and process throughout the Hadean and into the Archean, are both important contributions to the ongoing debate about the formation of Earth’s earliest stable crust.

I’m grateful to have had the opportunity to review this manuscript. It was thought provoking and I learned a lot. As the authors say in their introduction, studies of the Hadean are “fraught with controversy,” and I read the manuscript through that lens. Below I offer several comments and suggestions. Some of my suggestions may sound big (I think more discussion about, and evaluation of, the new partition coefficients is needed!), but they’re all offered in the spirit of “accept with minor revisions.” I hope that the authors’ response to some or all of my comments will help make the manuscript even more compelling to a critical reader of Nature Communications. I’m certainly happy to take another look at a manuscript before it goes to press, should that be useful for the authors and/or the editor.

General comment:

Try to use “calculated melt” or “model melt” consistently throughout (instead of just “melt”) when referring to compositions determined using partition coefficients or proxies or fractionation factors.

This has been done.

Line-by-line comments:

Line 19: Is it possible to distinguish between continental arcs and oceanic arcs using this dataset? (line 141 suggests you're thinking about island arcs)

We have rehailed our partition coefficients and our discrimination diagrams by plotting a larger dataset of arc lavas and we have separately plotted island arcs and continental arcs. While there are some datapoints that fall exclusively in the island arc lava field, most of them are in overlapping regions of the island arc and continental arc fields.

Line 45: I think a reference to Ackerson et al. (2021) would be appropriate and useful here.

This has been added.

Line 45: I kept my eyes open for this discussion of Al-in-zircon because I'm pretty interested in it now, but I didn't find it again in the main text until line 172, in the concluding paragraph. Figures S2A and S2B are interesting and thought provoking, but given the prominent mention of Al in the intro and conclusion, I'm surprised there's not more discussion of Al-in-zircon (and peraluminous/metaluminous melts) in the main text. If there's not enough space in the main text to add a few more words about this, please refer explicitly to Figure S2 in the text (if journal rules permit it) and expand the caption.

The Al measurements have been briefly discussed in the main section.

Line 50: "To derive the SiO₂ content and Si+O isotopic values of our zircons' parental melts, we first model their parental melt trace element chemistry

This has been fixed.

Line 53 / Line 68 / Online Methods Line 197: Is this discussion of GEOROC describing what was done by Turner et al. (2020), or new work you have done to test or refine the Th/Y proxy? If reporting previous work, I think the parenthetical explanation at line 53 can be cut for clarity (let the reference do the explaining). If new, please provide the search criteria and access date used to assemble the whole-rock dataset from GEOROC. I'd also be curious to know the logic behind using whole-rock data instead of volcanic glass and melt inclusions, which might be more relevant for constraining melt composition at the time of zircon crystallization. If there is no difference between Th/Y vs SiO₂ in whole rock compositions and in glass compositions, that would be interesting and important to know, too.

It is Turner's work and the line has been fixed.

"Then, using melt Th/Y vs. SiO₂ content calibration of [6], we calculate the wt% SiO₂ concentration of JHZ parent melts."

Line 59: "zircon analyzed" or zircon analysis? Did any crystals have multiple analytical spots?

The latter. Some crystals had multiple analysis locations.

Line 62: the parenthetical discussion of serpentinites is a little distracting here—can it be moved to a relevant figure caption, or elsewhere?

This statement serves as an argument for the novelty of this study. So, we have decided to keep it in here.

Line 73: I appreciate the importance of using partition coefficients tailored to your specific system. This is something I've thought a lot about, and I'm glad others are thinking about it, too! However, it's not clear to me why/how your D values are an improvement over other published D values. Why are they better suited for the JH zircons than other published partition coefficients out there. Should we all be using these new D values when we calculate model melts for the JH? If so, why? Tell us clearly! I like that you "wanted to evaluate whether temperature, pressure, water, and alkali-content of the melt could yield different D values" (online methods line 138-140), but it's not clear to me that you varied any parameters other than temperature (1100 vs 1300 C). The significance of the different D values at different temperatures would be useful to discuss, as would the similarities to the Burnham and Berry (2012) Ds, and the differences with the Padilla and Gualda(2016) Ds. It's possible I lost the thread as I moved back and forth between the main text, the online methods, supplemental table 1, supplemental table 5, and supplemental figure 5 trying to piece this together. Having a more centralized/streamlined presentation, evaluation, and discussion of these new partition coefficients would be very useful for readers like me who are excited by this work and want to follow the logic of it (in the supplement is fine). Claiborne et al. (2018) and Burnham (2020) have interesting discussions about environmental factors (e.g., temperature) influencing the partitioning of REE in zircon. It might be worthwhile to cite one or both of these papers in a future discussion.

Thanks for this comment. We now present a more robust treatment of the partition coefficients in this revised manuscript. With new experimental data, we generated $\log D$ vs $10^4/T(K)$ calibrations. We do see differences between the D-values derived at JHZ crystallization temperatures and those of previously published experimental studies. There are also differences between our D-values and those reported by Claiborne et al., 2018 and Padilla and Gualda, 2016 (P&G) from empirical datasets. However, we do not think that these are directly comparable since our experiments deal with intermediate melts while the studies just mentioned analyze highly silicic melts (~75%). There is also a greater chance of inclusions corrupting the zircon measurements in natural zircons. Moreover, Fig. S5 shows the comparison and differences between our calibration and the four studies mentioned in this comment. While we use our new experimental data to derive the appropriate D-values for our calculations, for transparency we have also performed a regression after combining our experimental dataset with that of Claiborne and P&G. The latter does change the derived D-values, however, the modeled JHZ melt compositions still fall in the arc fields so our interpretation remain unaffected.

Line 75: The D values determined at 1100 C? Why did you choose these instead of the ones determined at 1300 C? (I can guess, but your evaluation would carry more weight)

See above

Line 76: I'm very curious, would your model melt SiO₂ compositions have been substantially different (enough to alter your final interpretations) if you had calculated Th/Y ratios determined using different partition coefficients? (the higher T ones you calculated, plus the B+B and P+G ones used in your Fig S5, maybe also Sano et al., 2002 and Claiborne et al., 2018)? Interesting if the answer is yes, interesting if the answer is no. It would be great to know!

Considering the new discrimination diagrams, it does not seem to change the final interpretation.

Line 107: Reference 21 (Savage et al., 2011)—is this paper (largely focused on Hekla, Iceland) the one you meant to cite here?

No. This mistake has been rectified

Line 119-120 / Figure 3A: Clarify—is the d_{30Si} mantle value the same as the MORB reference value?

Clarified.

Line 141: Carley et al. (2022) came to a similar conclusion, but reported greater similarities with modern continental arcs than modern island arcs.

Based on our data, we do not think a distinction may be drawn.

Line 142 + 145: include the word “calculated” or “model” to describe the JHZ melt

It has been done.

Line 146: Clarify—Turner et al. (2020) used different D and also different zircon crystals and data?

This has been changed to *“With the melt TE/REE values derived using our new partition coefficients, we compared modeled JHZ melts and melt values derived from zircon values of [6] to lavas from modern regimes where crust is generated on Earth, on discrimination diagrams (Fig. 4 A&B).”*

Line 153-155: I think a reference to Davidson et al. 2007 and/or 2013 would be appropriate and useful

Davidson et al., 2013 has been added.

Line 161-164: continue to emphasize “calculated” melt.

It has been fixed

Line 161-164: “one” and “a couple” and “the majority” are a little unsatisfying without being reminded of how many calculated melts you have in total. Anchor your reader with some real numbers.

This has been fixed.

Figure comments:

Figure 1: Explain in the caption, what is the significance of 720C? Why is this the dividing line for differently colored symbols? Why not (for example) the average temperature of 685 that you told us about in the text?

It has been done.

“The zircon population is divided at 720 °C because this is the upper limit of the “wet” granite solidus.”

Figure 1: Consider adding a dashed line (or similar) to show the Hadean-Archean boundary (important to clearly establish the age you use for the next figures).

We have decided to combine the two zircon populations as per the another reviewer's suggestion.

Figure 1: It would be interesting to add Iceland in addition to MORB and OIB (same comment applies to Figure 4)

We tried this but it adds so much information as to clutter both figures; because we do not talk about Icelandic magmas in the manuscript, we feel the figures are more instructive without this data.

Figure 1: please add n-values to caption or legend

It has been done

Figure 1 caption: The difference in model melt SiO₂ is interesting (and exciting!), but I need to know, why are your D values better suited to this effort than the ones presented by Turner et al. (2020) who used Burnham and Berry (2012)? I'm ready to be persuaded!

We have used the same D values for our JHZs as well as Turner et al.'s JHZs. With the new D values, our melt SiO₂ wt. % average is slightly lower than that of Turner's. We feel the new D values are more appropriate because these are extrapolations based on experimental calibrations. Moreover, these experiments were done at 1 GPa using a metaluminous to peraluminous granitic system (as opposed to Burnham and Berry, 2012. who conducted zircon-melt partitioning at 1 atm in a system without alkalis).

Figures 1 and 4: I'm curious—if you used Turner et al. (2020) data and your D values, or your data with the Burnham and Berry (2012) D values used by Turner et al. (2020), would trends change substantially? If this is what you've already done, please clarify!

The former. We have used our data+Turner's data with our D values. We think that there is a significant change if we opted for the alternative as shown below.

Figure 2: This is a very data-rich figure! It's overwhelming to untangle what is in the legend, what is labeled directly, and what is named/described in the caption. There seems to be a lot going on in the olive green box, and it's hard to spot. The two circular symbols below the horizontal MORB line are unclear/confusing--are these fields for the serpentinites? Please try to streamline (maybe more labels? Maybe more in the legend?)

This figure has been more streamlined.

Figure 3A: Is the chert reference in the legend intentionally? Seems out of place. '

Fig 3 has been altered for clarity.

Figure 3A: Make clearer in the legend (not just the main text at line 118) the difference between the green-box chert and the gray-box chert.

All cherts have now been coloured similarly.

Figure 3 (A+B): Add n-values to legend or caption

It has been done.

References in this document that are not in the main text

Anastassia Y. Borisova, Anne Nédélec, Nail R. Zagrtednov, Michael J. Toplis, Wendy A. Bohrsen, Oleg G. Safonov, Ilya N. Bindeman, Oleg E. Melnik, Gleb S. Pokrovski, Georges Ceuleneer, Klaus Peter Jochum, Brigitte Stoll, Ulrike Weis, Andrew Y. Bychkov, Andrey A. Gurenko; Hadean zircon formed due to hydrated ultramafic protocrust melting. *Geology*, **50** (3): 300–304 (2022).

Drabon, N., Byerly, B.L., Byerly, G.R., Wooden, J.L., Keller, C.B. and Lowe, D.R., Heterogeneous Hadean crust with ambient mantle affinity recorded in detrital zircons of the Green Sandstone Bed, South Africa. *Proceedings of the National Academy of Sciences*, **118**(8), (2021).

Grimes, C.B., John, B.E., Kelemen, P.B., Mazdab, F.K., Wooden, J.L., Cheadle, M.J., Hanghøj, K. and Schwartz, J.J. Trace element chemistry of zircons from oceanic crust: A method for distinguishing detrital zircon provenance. *Geology*, **35**(7), 643-646 (2007).

Harrison, T.M., Bell, E.A. and Boehnke, P. Hadean zircon petrochronology. *Reviews in Mineralogy and Geochemistry*, **83**(1), pp.329-363, (2017).

J. Elis Hoffmann, Chao Zhang, Jean-Francois Moyen, Thorsten J. Nagel, Chapter 7 - The Formation of Tonalites–Trondjemite–Granodiorites in Early Continental Crust, Editor(s): Martin J. Van Kranendonk, Vickie C. Bennett, J. Elis Hoffmann, *Earth's Oldest Rocks (Second Edition)*, Elsevier, 2019, Pages 133-168, ISBN 9780444639011, <https://doi.org/10.1016/B978-0-444-63901-1.00007-1>.

Reviewer #1 (Remarks to the Author):

I am pleased to see that the authors have made a significant effort to address my comments. The only suggestion that they have not followed is to explicitly state that their approach is an expansion of a previously published methodology (their reference 6) that I think should be made clearly in the first few sentences of the paper. This does not detract from their work but acknowledges a precursor study. This is important not just to give fair credit but because this is only one of several studies I am aware of that have followed reference 6 in inverting the zircon trace elements to get melt compositions. The authors can then clearly state that what their new advances are (e.g. new D's, Si and O isotopes and identification of recycled components in the source). If that final change is made I would be happy to recommend publication of the revised version.

Reviewer #3 (Remarks to the Author):

In "Eoarchean/Hadean melts reveal arc-like trace element and isotopic chemistry," Chowdhury and colleagues use zircon from the Jack Hills of Western Australia to expand our understanding of Earth's earliest crust-forming processes. They conclude that a subduction setting, with complicated water-rock interactions and inputs of detrital sediments, was needed to generate the magmas from which Jack Hills zircon crystallized. I served as Reviewer #3 for the initial submission of this article to Nature Communications. In my initial review, I requested a more robust treatment of partition coefficients, pointed out a few areas that could use clarification, and made other small suggestions for improvement. In their revised manuscript and rebuttal document, Chowdhury and colleagues appropriately addressed my suggestions, or defended their decision to disregard some of my comments; I think the same is true of their treatment of comments by the other two reviewers. I am satisfied with the revised manuscript and support its publication in Nature Communications; it is an exciting addition to the field of Earth Earth studies, and one that will be useful and thought provoking for readers. I am happy to review the manuscript again in the future should the other reviewers recommend it be returned to the authors for another round of revisions, though I fully expect to give the same assessment at that time.

REVIEWER COMMENTS AND RESPONSES

Reviewer #1 (Remarks to the Author):

I am pleased to see that the authors have made a significant effort to address my comments. The only suggestion that they have not followed is to explicitly state that their approach is an expansion of a previously published methodology (their reference 6) that I think should be made clearly in the first few sentences of the paper. This does not detract from their work but acknowledges a precursor study. This is important not just to give fair credit but because this is only one of several studies I am aware of that have followed reference 6 in inverting the zircon trace elements to get melt compositions. The authors can then clearly state that what their new advances are (e.g. new D's, Si and O isotopes and identification of recycled components in the source). If that final change is made I would be happy to recommend publication of the revised version.

We have made the suggested change:

L42-48: “We use in-situ analyses of Jack Hills Zircons (JHZs), new zircon-melt partition coefficients, and zircon-melt Si+O isotope fractionation factors to place quantitative constraints on the chemistry of their parental magmas. The inversion of zircon chemistry to derive parent melt chemistry follows the methodology developed by [6], and we further expand upon that technique by utilizing a new suite of partition coefficients to quantify trace element (TE) and rare-earth (REE) element concentrations, and SiO₂ content of the zircon parental melts. Using this information, we then derive the Si+O isotopic values of the Eoarchean and Hadean melts as well as their formational regimes.”

Thank you for the suggestion.

Reviewer #3 (Remarks to the Author):

In “Eoarchean/Hadean melts reveal arc-like trace element and isotopic chemistry,” Chowdhury and colleagues use zircon from the Jack Hills of Western Australia to expand our understanding of Earth’s earliest crust-forming processes. They conclude that a subduction setting, with complicated water-rock interactions and inputs of detrital sediments, was needed to generate the magmas from which Jack Hills zircon crystallized. I served as Reviewer #3 for the initial submission of this article to Nature Communications. In my initial review, I requested a more robust treatment of partition coefficients, pointed out a few areas that could use clarification, and made other small suggestions for improvement. In their revised manuscript and rebuttal document, Chowdhury and colleagues appropriately addressed my suggestions, or defended their decision to disregard some of my comments; I think the same is true of their treatment of comments by the other two reviewers. I am satisfied with the revised manuscript and support its publication in Nature Communications; it is an exciting addition to the field of Earth Earth studies, and one that will be useful and thought provoking for readers. I am happy to review the manuscript again in the future should the other reviewers recommend it be returned to the authors for another round of revisions, though I fully expect to give the same assessment at that time.

Thank you for your considerable input that has improved the quality of our manuscript.